# Spectral-power associations reflect amplitude modulation and within-frequency interactions on the sub-second timescale and cross-frequency interactions on the seconds timescale

**Melisa Menceloglu[1], Marcia Grabowecky[1,2], Satoru Suzuki** [1,2]*

**1** Department of Psychology, Northwestern university, Evanston, Illinois, United States of America,
**2** Interdepartmental Neuroscience, Northwestern University, Evanston, Illinois, United States of America

* satoru@northwestern.edu

**Data Availability Statement:** Relevant data are in the paper and its Supporting Information files.

## Abstract

We investigated the global structure of intrinsic cross-frequency dynamics by systematically examining power-based temporal associations among a broad range of oscillation frequencies both within and across EEG-based current sources (sites). We focused on power-based associations that could reveal unique timescale dependence independently of interacting frequencies. Large spectral-power fluctuations across all sites occurred at two characteristic timescales, sub-second and seconds, yielding distinct patterns of cross-frequency associations. On the fast sub-second timescale, within-site (local) associations were consistently between pairs of $\beta$—$\gamma$ frequencies differing by a constant $\Delta f$ (particularly $\Delta f \sim$ 10 Hz at posterior sites and $\Delta f \sim$ 16 Hz at lateral sites) suggesting that higher-frequency oscillations are organized into $\Delta f$ amplitude-modulated packets, whereas cross-site (long-distance) associations were all within-frequency (particularly in the >30 Hz and 6–12 Hz ranges, suggestive of feedforward and feedback interactions). On the slower seconds timescale, within-site (local) associations were characterized by a broad range of frequencies selectively associated with ~10 Hz at posterior sites and associations among higher (>20 Hz) frequencies at lateral sites, whereas cross-site (long-distance) associations were characterized by a broad range of frequencies at posterior sites selectively associated with ~10 Hz at other sites, associations among higher (>20 Hz) frequencies among lateral and anterior sites, and prevalent associations at ~10 Hz. Regardless of timescale, within-site (local) cross-frequency associations were weak at anterior sites indicative of frequency-specific operations. Overall, these results suggest that the fast sub-second-timescale coordination of spectral power is limited to local amplitude modulation and insulated within-frequency long-distance interactions (likely feedforward and feedback interactions), while characteristic patterns of cross-frequency interactions emerge on the slower seconds timescale. The results also suggest that the occipital $\alpha$ oscillations play a role in organizing higher-frequency oscillations into ~10 Hz amplitude-modulated packets to communicate with other regions. Functional implications of these timescale-dependent cross-frequency associations await future investigations.

Additionally, the EEG data are deposited at Dryad under doi: 10.5061/dryad.hdr7sqvf9.

**Funding:** Yes. NIH T32 grant to MM.

**Competing interests:** The authors have declared that no competing interests exist.

## Introduction

Because oscillatory dynamics are prevalent in the brain (e.g., [1]), many EEG, MEG, and Electrocorticography (ECoG) studies have examined the roles of oscillatory neural activity in perception (e.g., [2–5]), attention (e.g., [6–9]), memory (e.g., [10]), cognition (e.g., [11–14]), and the general control of neural communications (e.g., [15]). Some studies examined the dynamics of oscillatory activity while people rested with their eyes closed to identify intrinsic spatial networks of oscillatory activity (e.g., [16–19]) that may correlate with individual differences (e.g., [20]) or neural dysfunctions (e.g., [21–23]). The strategies used by most prior studies were to examine phase-phase, power-power, and/or phase-amplitude associations within specific frequency bands or specific combinations of frequency bands to identify spatial networks mediated by those bands.

The goal of the current study was complementary in that instead of identifying spatial networks based on associations within specific frequency bands, we examined the global distribution of cross-frequency associations, focusing on both local characteristics reflected in cross-frequency associations within individual EEG-derived current sources (which we call sites here) and long-distance characteristics reflected in cross-frequency associations between sites. Importantly, we observed that spectral power universally (across oscillation frequencies and sites) fluctuated on two distinct timescales, sub-second and seconds. Because these fast and slow fluctuations are mathematically orthogonal (see below), distinct cross-frequency mechanisms may operate on these timescales, potentially yielding timescale-dependent patterns of cross-frequency associations at local and long-distance levels. We thus examined power-power associations that can independently transpire on different timescales; that is, slow cross-frequency co-variations in spectral power may occur irrespective of the presence of fast co-variations, and vice versa. In contrast, the timescale of a phase-phase or phase-amplitude coupling is constrained by the interacting oscillation frequencies. Analyses of spectral-power associations (though less temporally precise than analyses of phase coupling) provide additional advantages for characterizing the overarching structure of cross-frequency interactions.

First, spatial patterns of spectral-power associations may reveal the distribution of particularly consequential cross-frequency interactions because adjusting phase coupling would likely have more impact for coordinating interactions between oscillatory activities that are concurrently elevated in power, thereby influencing larger neural populations (because larger spectral power implies larger synchronously oscillating populations). Power-based associations may also reveal interactions that are phase insensitive; for example, synchronized oscillatory output from one population may excite another population into generating oscillatory activity at a different frequency. Second, analyses of spectral-power associations may reveal the global distribution of phase-amplitude couplings. Specifically, consistent covariations of power between pairs of oscillation frequencies differing by a constant $\Delta f$ at a given current source imply the presence of beating at $\Delta f$, that is, the presence of oscillation at the pair-average frequency, $\bar{f}$ Hz, being amplitude-modulated at $\Delta f$ Hz. This in turn implies phase-amplitude coupling because the periodic $\Delta f$ Hz modulation of the amplitude (power) of a $\bar{f}$ Hz oscillation is likely mediated by its interaction with the phase of a $\Delta f$ Hz oscillation.

To facilitate straightforward interpretations of our results, we applied only minimal and necessary data transformations: (1) taking the temporal derivative of EEG to reduce the non-oscillatory background spectra, (2) applying the surface-Laplacian transform to localize neural sources in a purely data-driven manner and to reduce volume conduction and electrode-referencing effects, and (3) applying Morlet-wavelet convolution to extract spectral power as a function of time.

Overall, the results have revealed surprisingly simple local and long-distance patterns of intrinsic spectral-power associations characterized by oblique (parallel to diagonal), diagonal, and columnar patterns depending on timescale (sub-second or seconds), region (posterior, central, lateral, vs. anterior) and spatial scale (within vs. across EEG-derived current sources). These patterns of spectral-power associations provide a reference for understanding how intrinsic cross-frequency dynamics adjust to behavioral demands and sensory dynamics. Further, while recent studies have demonstrated the relevance of $\alpha$ rhythm in visual perception and its association with occipital $\alpha$ oscillations, the current results suggest that the occipital $\alpha$ oscillations play a role in organizing higher-frequency oscillations into ~10 Hz amplitude-modulated packets to communicate with other areas.

## Materials and methods

### Participants

Twenty-four Northwestern University students (15 women, 1 non-binary; ages 18 to 24 years, $M = 21.25$, $SD = 1.57$) gave informed written consent to participate for monetary compensation ($10/hr). All were right-handed, had normal hearing and normal or corrected-to-normal vision, and no history of concussion. They were tested individually in a dimly lit room. The study protocol (including the consent procedure) was approved by the Northwestern University Institutional Review Board.

### EEG recording and pre-processing

While participants engaged in spontaneous thought with their eyes closed for approximately 5 minutes, EEG signals were recorded from 60 scalp electrodes (although we used a 64-electrode montage, we excluded EEG signals from noise-prone electrodes, *Fpz*, *Iz*, *T9*, and *T10* from analyses) at a sampling rate of 512 Hz using a BioSemi ActiveTwo system (see www.biosemi.com for details). Two additional electrodes were placed on the left and right mastoid area. The EEG data were preprocessed using EEGLAB and ERPLAB toolboxes for MATLAB [24,25]. The data were re-referenced offline to the average of the two mastoid electrodes, bandpass-filtered at 0.01 Hz-80 Hz, and notch-filtered at 60 Hz (to remove power-line noise that affected the EEG data from some participants). To reduce any effects of volume conduction and reference electrode choices, as well as to facilitate data-driven EEG current source discrimination, we applied the surface-Laplacian transform to all EEG data (e.g., [26–28]) using the Perrin and colleagues' method (e.g., [29–31]) with a typical set of parameter values (e.g., [32]).

### EEG analysis

**The use of EEG temporal derivative.** We refer to the surface-Laplacian transformed EEG signals simply as EEG signals. In all figures, whenever units are not shown, they are arbitrary (a.u.), and the shaded regions indicate ±1 standard error of the mean. An example 1 s EEG waveform at a central site *FCz* from one participant is shown in Fig 1A (black curve). The waveform reflects the synchronized electrical activity from a large neural population that generates a distinct current source at *FCz*. The mean spectral-amplitude profile (i.e., fast Fourier transform, FFT) computed on 59 consecutive 5 s waveforms for the same participant is shown in Fig 1B (black curve; the shaded region represents ±1 standard error of the mean). The general linear decrease in the spectral amplitude for higher frequencies with a slope of approximately 1 (in log-log scale) reflects the $1/\Delta f$ background profile largely explained by the neuronal Ornstein-Uhlenbeck process that exhibits a random-walk type behavior (e.g., [33,34]; see [35] for a review of the various factors that contribute to the $1/f^\beta$ background). The

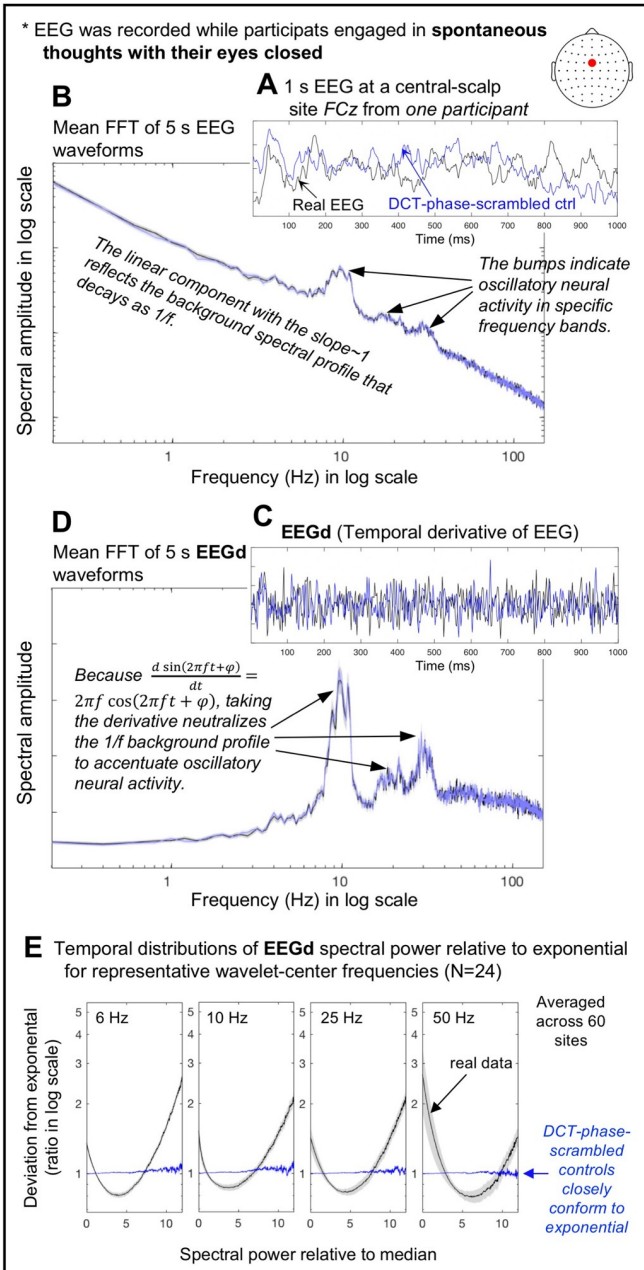

**Fig 1. The use of the temporal derivative of EEG, EEGd, and DCT-phase-scrambled controls for examining intrinsic cross-frequency spectral-power associations. A.** An example of a 1 s EEG waveform (black) and its DCT-phase-scrambled control (blue) at *FCz* from one participant. **B.** Mean fast Fourier transforms (FFTs) computed on the fifty-nine 5 s waveforms (same site and participant) for the real EEG (black) and its DCT-phase-scrambled control (blue) plotted in a log-log format. **C.** The temporal derivatives, which we call EEGd, of the example EEG waveform (black) and its DCT-phase-scrambled control (blue) shown in A. **D.** Mean FFTs computed on the fifty-nine 5 s waveforms (same site and participant) for the real EEGd (black) and its DCT-phase-scrambled control (blue) plotted in a semi-log format. For B and D, the shaded regions represent ±1 standard error of the mean based on the FFTs computed on multiple 5 s waveforms. **E.** Temporal distributions of EEGd spectral power for representative wavelet-center frequencies (6, 10, 25, and 50 Hz) for the real data (black) and their DCT-phase-scrambled controls (blue), plotted as deviations from exponential fits to the DCT-phase-scrambled controls (averaged across all sites). The deviations are shown as ratios in log scale, and spectral power (*x*-axis) has been normalized to the median (per frequency per site per participant). The shaded regions represent ±1 standard error of the mean with participants as the random effect. Note that the DCT-phase-scrambled data provide suitable stochastic controls because their exponential spectral-power distributions imply a Poisson point process.

spectral "bumps" seen around 10 Hz, 20 Hz, and 30 Hz indicate the characteristic bands of oscillation frequencies that the neural population reflected at this site for this person may utilize for neural communication and/or information processing. Taking the temporal derivative of EEG ($\frac{\Delta EEG}{\Delta t}$, where $t$ is the temporal resolution, i.e., 1/512 s; see black curve in Fig 1C) largely removes the $1/\Delta f$ background (due to trigonometric properties) to highlight the oscillation bumps (see black curve in Fig 1D). While Fig 1D shows an example at one site from one participant, we confirmed that the background spectral-amplitude profiles were generally flattened after taking the temporal derivative. Because we wished to investigate the dynamics of oscillatory activity (over and above the general $1/f^\beta$ background), we used the EEG temporal derivative, which we call EEGd.

**Computing spectral power as a function of time.** The spectral-amplitude profiles shown in Fig 1B and 1D are time-averaged (standard fast Fourier transforms). As we wished to understand how the brain dynamically coordinates oscillatory activity, we used a Morlet wavelet-convolution method (e.g., [32]) to extract spectral amplitudes as a function of time to investigate how spectral power (amplitude squared) fluctuated over time. Specifically, each EEGd waveform was decomposed into a time series of spectral power using 200 Morlet wavelets with their center frequencies, $f_c$'s, and factor $n$'s (roughly the number of cycles per wavelet, which is related to the temporal standard deviation of the wavelet by, $n = 2\pi f \cdot SD$), both logarithmically spaced (because neural temporal-frequency tunings tend to be approximately logarithmically scaled; e.g., [36,37]). The $f_c$'s spanned the range of 3 Hz to 60 Hz and the $n$'s spanned the range of 3 to 16, resulting in the temporal resolutions of $SD$ = 159 ms (at 3 Hz) to $SD$ = 42 ms (at 60 Hz) and spectral resolutions of $FWHM$ (full width at half maximum) = 2.36 Hz (at 3 Hz) to $FWHM$ = 8.83 Hz (at 60 Hz). These values struck a good balance for the temporal/spectral-resolution trade-off, and are typically used in the literature (e.g., [32]).

**Generating phase-scrambled controls.** To remove spurious cross-frequency associations between proximate $f_c$'s due to partial spectral overlaps of Morlet wavelets, we generated phase-scrambled controls whose spectral power fluctuated stochastically (i.e., unpredictably in a memory free manner) while their time-averaged spectral-amplitude profiles matched those of the real data. While phase-scrambling can be performed using several different methods, we chose discrete cosine transform, DCT (e.g., [38]). In short, we transformed each 5 min EEG waveform with type-2 DCT, randomly shuffled the signs of the coefficients, and then inverse-transformed it with type-3 DCT, which yielded a phase-scrambled version. DCT phase-scrambling is similar to DFT (discrete Fourier transform) phase scrambling except that it is less susceptible to edge effects. We verified that DCT phase-scrambling yielded a desired outcome, generating waveforms whose spectral-power fluctuations conformed to exponential distributions (Fig 1E) indicative of a Poisson point process (i.e., a stochastic process), with virtually no distortions to the time-averaged spectral-amplitude profiles of EEG or EEGd (e.g., the blue and black curves overlap in Fig 1B and 1D). In passing, it is noteworthy that the distributions of spectral-power fluctuations of the real data deviated from exponential in a U-shaped manner with disproportionately frequent occurrences of near-zero and high values (Fig 1E), indicative of intermittent bursts (relative to stochastic variations).

We used EEGd from 60 sites in conjunction with their DCT-phase-scrambled controls to extract intrinsic spectral-power associations over and above any artifact due to partial wavelet overlaps. Because of the 5 min length of the data (containing 153,600 time points at the 512 Hz sampling rate), 200 wavelet-center frequencies, and 60 sites per participant, even one set of phase-scrambled data would be unlikely to generate accidental patterns of spectral-power associations. We verified that the control patterns of spectral-power associations generated with different versions of phase-scrambled data (using different random seeds) were virtually

indistinguishable, especially when two or more control patterns were averaged. To be conservative, we generated four versions of phase-scrambled data to compute four sets of control spectral-power associations (per site per participant), and used their averages as the controls (see below). We are thus reasonably certain that the control patterns accurately captured the spurious associations due to partial wavelet overlap.

**Computing spectral-power associations.** For assessing power-based cross-frequency coupling, we chose not to use a conventional correlation measure. One reason is that the stochastic distribution of EEGd spectral power is not normal but exponential (Fig 1E). The other reason is that it is difficult to remove artifacts due to wavelet overlap in the frequency domain using a conventional correlation measure (see below). Instead, we computed *top-bottom-percentile based* temporal associations. This method has advantages including (1) not being susceptible to outliers and (2) the straightforwardness of the computed association values that directly indicate the log power modulation in frequency *Y* that is temporally associated with statistically large power variations in frequency *X*. We computed spectral-power associations at individual scalp-based current sources—*within-site* associations—as well as between pairs of current sources—*cross-site* associations.

**Within-site (local) spectral-power associations.** To evaluate a within-site spectral-power association from a probe frequency, $f_{\text{probe}}$, to a test frequency, $f_{\text{test}}$, we first identified the time points corresponding to the top and bottom 15% in the $f_{\text{probe}}$ power time-series. This criterion was reasonable for an approximate exponential distribution (Fig 1E), capturing a good proportion (accounting for approximately 70% of the total variability in the sum of squares) of statistically large variations. The $f_{\text{test}}$ power was averaged separately over the time points corresponding to the top-15% and bottom-15% for the $f_{\text{probe}}$ power time-series, and the strength of $f_{\text{probe}}$-to-$f_{\text{test}}$ spectral-power association was computed as the natural-log (Ln) ratio of these averages (thus normalizing for absolute power). Thus, a larger positive association value indicates that the top/bottom-15% variation in the probe-frequency power was associated with greater *same-signed* co-variation in the test-frequency power in Ln-ratio (analogous to positive correlation, indicating that the test-frequency power was higher when the probe-frequency power was in the top 15% than when it was in the bottom 15%). A larger negative association value indicates that the top/bottom-15% variation in the probe-frequency power was associated with greater *opposite-signed* co-variation in the test-frequency power in Ln-ratio (analogous to negative correlation, indicating that the test frequency power was lower when the probe-frequency power was in the top 15% than when it was in the bottom 15%). Association values near zero indicate the lack of consistent association (i.e., the top/bottom-15% power in the probe frequency tended to be equivalently coincident with higher or lower test-frequency power). In this way, we computed the strengths of probe-frequency-to-test-frequency spectral-power associations for all combinations of wavelet-center frequencies to generate a 2-dimensional, 200-probe-frequency-(*y*-axis)-by-200-test-frequency-(*x*-axis), association matrix per site per participant and averaged across participants. A cross-frequency association matrix computed in this way need not be symmetric about the diagonal. That is, *X*-Hz power modulation coincident with the top/bottom-15% *Y*-Hz power variation may be stronger or weaker than the *Y*-Hz power modulation coincident with the top/bottom-15% *X*-Hz power variation.

As mentioned above, because the wavelets used to extract spectral-power values had non-negligible spectral widths, proximate frequencies would appear to be associated due to partial wavelet overlap. This artifact would generate spurious positive associations near the diagonal (with the diagonal representing the Ln-ratio of the top-15% to bottom-15% power for each frequency). To remove this artifact, spectral-power associations were computed in the same way with the corresponding DCT-phase-scrambled data. To reliably estimate the artifact, we

generated control associations by averaging four association matrices computed using four different versions of phase-scrambled data (see above). We removed spurious associations due to partial wavelet overlap by taking the difference (in Ln-ratio) between the real and control associations (note that removing this artifact would not be as straightforward using a conventional correlation measure). This procedure also made the diagonal meaningful because values along the diagonal indicate the magnitudes of the top/bottom-15% variations in the real data relative to those expected by stochastic variation.

**Cross-site (long-distance) spectral-power associations.** Cross-site spectral-power associations were similarly computed. For example, to evaluate the spectral-power association from a probe frequency at site *S1*, *S1-f*$_\text{probe}$, to a test frequency at site *S2*, *S2-f*$_\text{test}$, we identified the time points corresponding to the top and bottom 15% in the *S1-f*$_\text{probe}$ power time-series. Then, the *S2-f*$_\text{test}$ power was averaged separately over the time points corresponding to the top-15% and bottom-15% for the *S1-f*$_\text{probe}$ power, and the strength of *S1-f*$_\text{probe}$-to-*S2-f*$_\text{test}$ spectral-power association was computed as the Ln-ratio of these averages. A larger positive association value indicates that the top/bottom-15% variation in the probe frequency power at *S1* was associated with greater *same-signed* co-variation in the test-frequency power at *S2* in Ln-ratio (analogous to positive correlation, indicating that the test-frequency power at *S2* was higher when the probe-frequency power at *S1* was in the top 15% than when it was in the bottom 15%). A larger negative association value indicates that the top/bottom-15% variation in the probe-frequency power at *S1* was associated with greater *opposite-signed* co-variation in the test-frequency power at *S2* in Ln-ratio (analogous to negative correlation, indicating that the test frequency power at *S2* was lower when the probe-frequency power at *S1* was in the top 15% than when it was in the bottom 15%). Association values near zero indicate the lack of consistent association (i.e., the top/bottom-15% power in the probe frequency at *S1* tended to be equivalently coincident with higher or lower test-frequency power at *S2*). In this way, we computed the strengths of *S1*-probe-frequency-to-*S2*-test-frequency spectral-power associations for all combinations of *S1* and *S2* wavelet-center frequencies to generate a 2-dimensional, 200-*S1*-probe-frequency-(*y*-axis)-by-200-*S2*-test-frequency (*x*-axis), association matrix per site-pair per participant and averaged across participants. Note that the diagonal indicates cross-site *within-frequency* spectral-power associations. The use of DCT-phase-scrambled controls was unnecessary in this case because partial wavelet overlap would not generate any spurious associations between data from separate sites.

The within-site and cross-site spectral-power associations were computed for the sub-second timescale (within a 500 ms interval) and the seconds timescale (across 500 ms intervals). The justification for examining spectral-power associations on these specific timescales is provided in the results section (see 1. Spectral power intrinsically fluctuates on two distinct timescales). Each 5 min EEG recording was divided into six-hundred 500 ms intervals. The sub-second timescale spectral-power associations were computed at a 512 Hz temporal resolution within each 500 ms interval and averaged across all intervals (per site or site-pair per participant). The seconds timescale associations were computed at a 500 ms resolution (with the data temporally averaged within each 500 ms interval) across the entire 5 min period (using each 500 ms interval as time unit). Thus, the sub-second-timescale and seconds-timescale spectral-power associations were mathematically independent.

## Results and discussion

### Spectral power intrinsically fluctuates on two distinct timescales

Half a second is a landmark timescale in that consciously accessible perceptual representations are thought to form through feedforward and feedback interactions within this timescale (e.g.,

[5,39,40]), and typical motor behavior such as spontaneous tapping, walking, and natural saccades have periodicities in this timescale (e.g., [41–43]). It is thus feasible that distinct neural interactions may transpire on (1) the *within-500-ms* timescale—potentially contributing to generating the basic building blocks of perceptual and motor processing—and (2) the *across-500-ms* timescale—potentially contributing to chaining these building blocks into cognition and action. Consistent with this reasoning, our data suggest that spectral power intrinsically fluctuates on these timescales.

Spectral-power fluctuations during a 500 ms interval from a posterior site, *PO7*, from one participant is shown in Fig 2A. As we focused on large (±15th percentile) variations (see the Methods section), the top-15% power is indicated in red, the bottom-15% power is indicated in blue, and intermediate power is shown as blank. This example clearly shows that spectral power goes through about 1–2 (red-blue) cycles per frequency within 500 ms. To confirm this observation across all 500 ms intervals for all sites and participants, we quantified the average duration of a top/bottom state (per 500 ms interval) as follows. For each frequency (i.e., for each wavelet-center frequency, $f_c$), we identified the state sequence; in the upper example shown in Fig 2A, the sequence is, *bottom-15% → in-between → top-15% → in-between → bottom-15% → in-between → top-15%*, or [**B–T–B–T**] where "–" indicates "in-between." We binarized the T/B states by ignoring the in-between states, and only counted states that began within the 500 ms interval of interest (to avoid redundant counting of states across 500 ms intervals). Thus, this example included 3 states, T, B and T, so that the estimated average duration of a T/B state for this frequency would be 500 ms/3 = 167 ms (Fig 2A). When the same (T or B) state recurred across an in-between state, we considered them to be a single state. In the lower example shown in Fig 2A, the identified sequence is, [**–B–B–T**], but the number of states would be 2, that is, a B followed by a T (combining the two B's across the in-between state), so that the estimated average duration of a T/B state for this frequency would be 500 ms/2 = 250 ms (Fig 2A). We computed the average T/B-state duration in this way for each of the six-hundred 500 ms intervals (over the 5 min EEG recording period) and averaged them across the 600 intervals per frequency per participant per site. We then generated a histogram of the average within-500-ms-interval T/B-state durations per site using the 4,800 values computed for the 200 frequencies and 24 participants; the histograms for the 60 sites are overlaid in Fig 2C. It is clear that the distributions of the average T/B-state durations were globally similar (i.e., similar across all 60 sites) and tightly peaked at ~230 ms with a negative tail that pulled the mean down to ~215 ms. This indicates that a 500 ms interval consistently contained a little over one cycle of substantial power fluctuation, confirming that 500 ms may provide an appropriate sub-second timescale to examine fast power fluctuations whose low/high states last ~230 ms.

An equivalent analysis was conducted for spectral-power fluctuations across 500 ms intervals over the 5 min period with the spectral power averaged within each 500 ms interval. The spectral-power-fluctuation data at *PO7* from the same participant is shown in Fig 2B with the time axis indicating the serial number of 500 ms intervals. This example clearly shows that the spectral power fluctuated between the top-15% (red) and bottom-15% (blue) states at a relatively stable rate. We computed the average T/B-state duration by dividing the entire duration (six-hundred 500-ms intervals) by the number of T/B states (counted in the same way as above) per frequency per participant per site. We generated a histogram of the average T/B-state durations per site using the 4,800 values computed for the 200 frequencies and 24 participants, and overlaid the histograms for the 60 sites (Fig 2D). It is clear that the distributions of the average T/B-state durations across 500 ms intervals were globally similar (i.e., similar across all 60 sites) and tightly peaked at about seven-and-a-half 500-ms intervals or ~3.75 s with a positive tail that pushed the mean up to ~5.44 s. Thus, slower spectral-power fluctuations occurred on the timescale of several seconds.

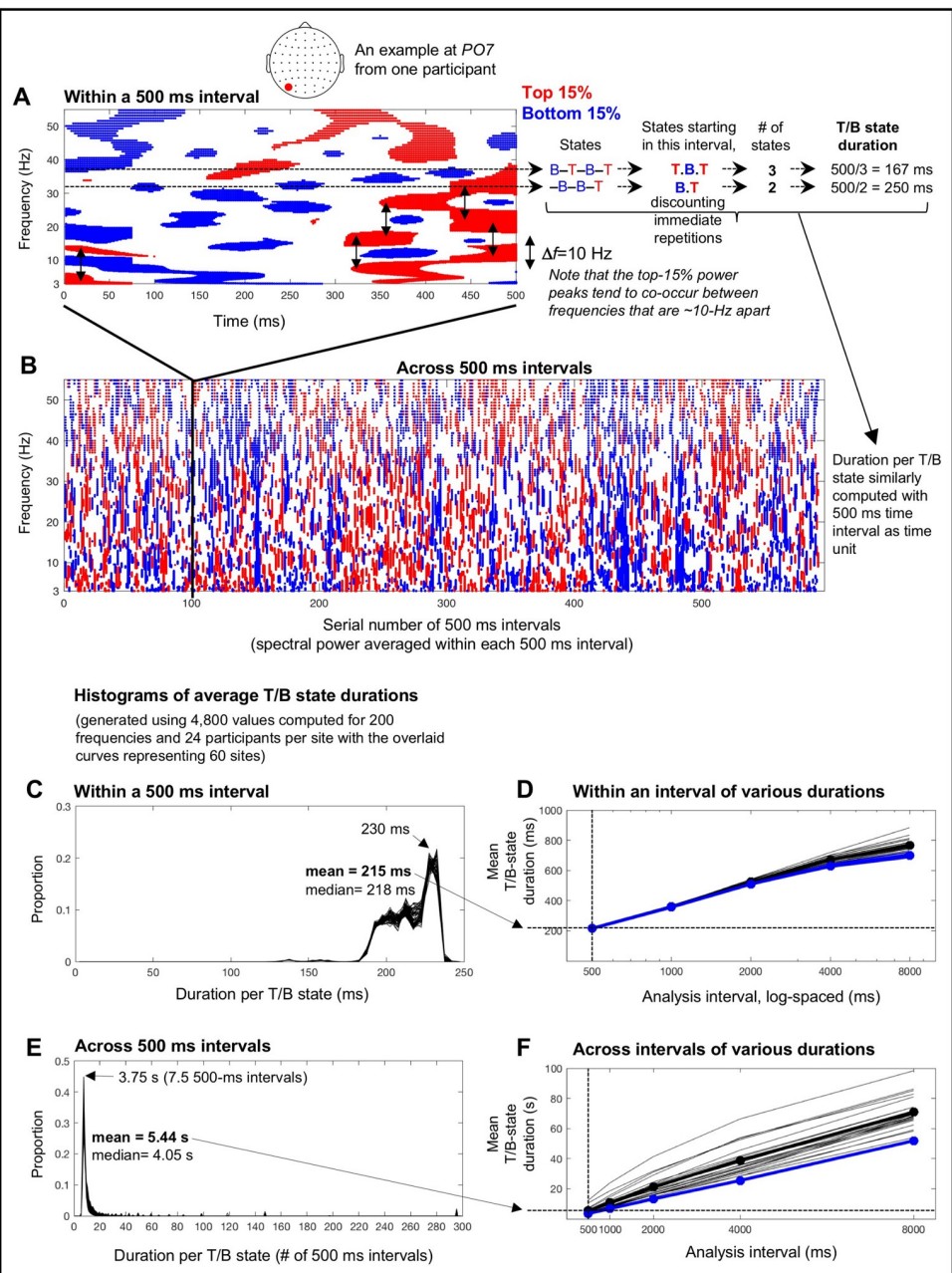

**Fig 2. Intrinsic spectral-power variations on the characteristic sub-second (~230 ms) and seconds (~3.75 s) timescales. A.** An example of the top-15% (red) and bottom-15% (blue) spectral-power variations within a 500 ms interval (taken from a 5 min EEG recording; see B) at a posterior site, *PO7*, from one participant. Note that spectral power tends to peak in pairs of frequencies that are ~10 Hz apart (the vertical arrows indicating *Δf*~ 10 Hz). The average duration of the top and bottom states was estimated per frequency by identifying the state sequence ("B" for bottom, "T" for top, and "–" for in-between states in the illustration), counting the top and bottom states that started within the interval while discounting immediate repetitions, and dividing the interval duration (500 ms in this case) by the number of T/B states. Two examples are shown for two different frequencies (see text for details). The average T/B-state duration computed for each of the six-hundred 500 ms intervals over the 5 min EEG recording period were averaged per frequency per participant per site. **B.** An example of the top-15% (red) and bottom-15% (blue) power variations occurring across 500 ms intervals (spectral power averaged within each 500 ms interval) over the 5 min period. The average duration of the top and bottom states was estimated in the same way as for the 500 ms interval (see A) but using 500 ms intervals as time unit. **C.** Histograms of average within-500-ms-interval T/B-state durations (averaged across 600 intervals) based on the 4,800 values computed for 200 wavelet-center frequencies and 24 participants per site; histograms from the 60 sites are overlaid. The mode of ~230 ms with the mean of ~215 ms

indicate that each brief high/low-power state typically lasted about a quarter of a second. **D.** Mean within-interval T/B-state durations similarly computed using different analysis intervals, 1000, 2000, 4000, and 8000 ms in addition to 500 ms, for both the real (black) and phase-scrambled (blue) data (the thin lines showing individual participants' data). The mean within-interval T/B-state duration approximately logarithmically increased with the length of analysis intervals. Note that the real and phase-scrambled curves virtually overlap, indicating that within-interval T/B-state durations depend primarily on spectral-amplitude compositions regardless of phase relations. Also note that individual differences are negligible for within-interval T/B-state durations computed with short (< 1000 ms) analysis intervals. **E.** Histograms of average across-500-ms-interval T/B-state durations based on the 4,800 values computed for 200 wavelet-center frequencies and 24 participants per site; histograms from the 60 sites are overlaid. The mode of ~3.75 s and the mean of 5.44 s indicate that each longer high/low-power state typically lasted several seconds. **F.** Mean across-interval T/B-state durations similarly computed using different analysis intervals (1000, 2000, 4000, and 8000 ms in addition to 500 ms) for both the real (black) and phase-scrambled (blue) data (the thin lines showing individual participants' data). The mean across-interval T/B-state duration approximately linearly increased with the length of analysis intervals. Note that the real and phase-scrambled curves nearly overlap for the 500 ms interval, indicating that T/B-state durations measured across 500 ms intervals primarily depend on spectral-amplitude compositions regardless of phase relations. Also note that individual differences are small for across-interval T/B-state durations computed with short (< 1000 ms) analysis intervals. We chose 500 ms as the analysis interval because it revealed the sub-second (within-interval) and seconds (across-interval) timescales of spectral-power fluctuations (1) that were relatively stable, depending on spectral-amplitude compositions regardless of phase relations, (2) that were highly consistent across participants, and (3) that appeared to support maximally distinct patterns of fast and slow spectral-power associations (see text for details).

Note that temporal estimates of the fast and slow fluctuations depend on the choice of the analysis interval. We chose 500 ms partly because it is a "landmark" timescale for human perception, cognition, and action (see above). The use of 500 ms analysis intervals also facilitated the goal of identifying distinctly timescale-dependent spectral-power associations.

Spectral-power fluctuations include both fast and slow components as shown in Fig 2A and 2B. Thus, estimates of within-interval T/B-state durations would necessarily increase with the use of longer analysis intervals because slower fluctuations would be included in the estimates. Indeed, the mean within-interval T/B-state duration approximately logarithmically increased with the length of analysis intervals (Fig 2D). The mean across-interval T/B state duration increased approximately linearly with the length of analysis intervals (Fig 2F). This is not surprising because we used non-overlapping intervals so that the within- and across-interval analyses of spectral-power dynamics were mathematically independent. The use of non-overlapping intervals made the temporal resolution for measuring across-interval power fluctuations inversely related to the length of the analysis interval.

Interestingly, both the within- and across-interval T/B-state durations were similar for the real (black curves) and phase-scrambled (blue curves) data for short (< 1000 ms) analysis intervals (Fig 2D and 2F), suggesting that the fast and slow spectral-power fluctuations revealed with short (< 1000 ms) analysis intervals primarily depend on spectral-amplitude compositions regardless of phase relations. It is also noteworthy that inter-participant variability was low (thin lines in Fig 2D and 2F) for both the fast (within-interval) and slow (across-interval) T/B-state durations obtained with short (< 1000 ms) analysis intervals, especially for those obtained with 500 ms intervals.

As shown below, the use of 500 ms analysis intervals revealed robust patterns of fast (within-interval) spectral-power associations. For the purpose of characterizing the timescale dependence of spectral-power associations, the use of the shortest analysis interval that generates distinct patterns of spectral-power associations would be advantageous for characterizing spectral-power associations on the fastest timescale because the use of longer intervals would intermix slower associations. For characterizing distinct spectral-power associations operating on the slower (across-interval) timescale that are mathematically independent of the fast (within-interval) associations, the use of 500 ms analysis intervals (relative to the use of longer analysis intervals) would also be most effective by providing the highest temporal resolution

for characterizing the slow across-interval dynamics. We verified this line of reasoning as follows.

We computed both within-interval and across-interval spectral-power associations for representative sites (for examining within-site spectral-power associations) and for representative site-pairs (for examining cross-site spectral-power associations), using 500, 1000, 2000, 4000, and 8000 ms analysis intervals. As shown below, the use of 500 ms analysis intervals revealed distinct patterns of fast (within-interval) and slow (across-interval) spectral power associations. Within-interval spectral-power associations computed with longer analysis intervals became progressively similar to the across-interval associations computed with a 500 ms analysis interval, confirming that the use of longer analysis intervals results in intermixing fast and slow associations. Across-interval spectral-power associations computed with longer analysis intervals largely maintained the patterns obtained with 500 ms analysis intervals except that the association magnitudes became weaker, confirming that the characteristic slow (across-interval) spectral-power associations are most effectively revealed with 500 ms analysis intervals (relative to longer analysis intervals).

In summary, the analyses presented in Fig 2 and the above observations suggest that the use of 500 ms analysis intervals reveals mathematically independent spectral-power dynamics on the sub-second (with high/low-power states typically lasting ~230 ms) and seconds (with high/low-power states typically lasting ~3.75 s) timescales with the following advantages: (1) these sub-second and seconds timescales are relatively stable, depending primarily on spectral-amplitude compositions regardless of phase relations, (2) they are highly consistent across participants, and most significantly (3) they appear to support maximally distinct patterns of spectral-power associations.

## Within-site (local) spectral-power associations on the (fast) sub-second timescale

The within-site spectral-power associations (in Ln-ratio relative to the phase-scrambled controls; see Materials and Methods/EEG analysis/Computing spectral-power associations) on the sub-second timescale (within a 500 ms interval) are marked by strong associations along the 45˚ lines parallel to the diagonal (see the 2D cross-frequency association plots in Fig 3A, 3B and 3D). This indicates that the associations are characterized by specific frequency differences, particularly, $\Delta f \sim$ 3 Hz, $\Delta f \sim$ 10 Hz, and $\Delta f \sim$ 16 Hz. The representative association patterns are shown in Fig 3A–3C. Posterior, lateral, and anterior sites yielded distinct patterns.

A representative posterior site shows systematic associations for the $\theta$-$\alpha$ frequencies along the $\Delta f \sim$ 3 Hz lines and particularly for the $\beta$-$\gamma$ frequencies along the $\Delta f \sim$ 10 Hz lines (e.g., Fig 3A, dashed ellipses). Note that the $\Delta f \sim$ 10 Hz associations can be seen within a single 500 ms interval as frequent co-occurrences of top-15% states in pairs of frequencies separated by ~10 Hz (e.g., vertical arrows in Fig 2A). We evaluated the inter-participant consistency of these patterns in three ways. One way was by plotting the association strength as a function of $\Delta f$ for each major frequency band (labeled $\theta$, $\alpha$, $\beta_1$, $\beta_2$, $\beta_3$, $\gamma_1$ and $\gamma_2$ in the figures). In this way, the associations for the $\theta$-$\alpha$ frequencies along the $\Delta f \sim$ 3 Hz lines (e.g., the main panel in Fig 3A) correspond to the $\theta$ and $\alpha$ curves both peaking at $\Delta f \sim$ 3 Hz (e.g., the upper-right panel in Fig 3A), while the associations for the $\beta$-$\gamma$ frequencies along the $\Delta f \sim$ 10 Hz lines (e.g., the main panel in Fig 3A) correspond to the $\beta_1$, $\beta_2$, $\beta_3$, $\gamma_1$ and $\gamma_2$ curves all peaking at $\Delta f \sim$ 10 Hz (e.g., the upper-right panel in Fig 3A). The tall peaks compared with the relatively small error regions (±1 standard error of the mean with participants as the random effect) indicate that the peaked associations along the $\Delta f \sim$ 3 Hz and $\Delta f \sim$ 10 Hz lines at posterior sites are consistent

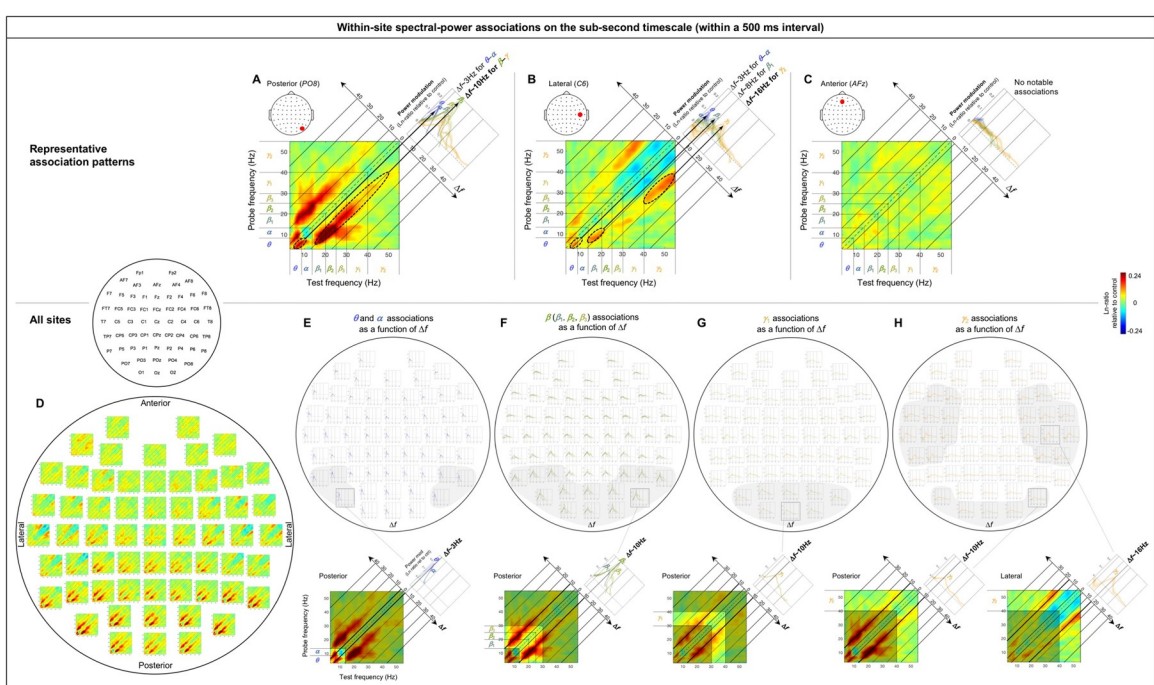

**Fig 3. Within-site spectral-power associations on the sub-second timescale (within a 500 ms interval).** The sub-second-timescale spectral-power associations were computed at 512 Hz temporal resolution within each 500 ms interval and averaged across all intervals (per site per participant) and averaged across participants. Each 2D-association plot shows the strengths of the same-signed (positive values, warmer colors) or opposite-signed (negative values, cooler colors) power modulation at all *test* frequencies (*x*-axis) that temporally coincided with the top/bottom-15% power variation at each *probe* frequency (*y*-axis), computed as Ln-ratio relative to the DCT-phase-scrambled control (to remove artifacts due to partial wavelet overlap; see text for details). Any asymmetry across the diagonal indicates directional differences; *X*-Hz power modulation coincident with the top/bottom-15% *Y*-Hz power variation may be stronger/weaker than the *Y*-Hz power modulation coincident with the top/bottom-15% *X*-Hz power variation. **A-C.** Representative association patterns. The strong positive associations along the 45° lines (A and B) indicate that spectral-power associations occurred at constant frequency differences, *f*'s. The accompanying line plots show the association strengths for major frequency bands as a function of *Δf* (warmer-colored curves representing higher-frequency bands), averaged across the *y*-side and *x*-side of the diagonal as the associations were approximately symmetric about the diagonal. The plots clearly show that the association strengths peaked at specific *Δf* values. **A.** An example from a representative posterior site (*PO8*) characterized by the *Δf* ~ 3 Hz associations for the $\theta$-$\alpha$ frequencies (small dashed ellipse) and particularly by the *Δf* ~ 10 Hz associations for the $\beta$-$\gamma$ frequencies (large dashed ellipse), suggesting that the $\theta$-$\alpha$ and $\beta$-$\gamma$ frequencies are amplitude-modulated at ~3 Hz and ~10 Hz, respectively, at posterior sites (see text for details). **B.** An example from a representative lateral site (*C6*) primarily characterized by the *Δf* ~ 16 Hz associations for the higher $\gamma$ frequencies (large dashed ellipse), suggesting that the higher $\gamma$ frequencies are amplitude-modulated at ~16 Hz at lateral sites. **C.** An example from a representative anterior site (*AFz*) characterized by the lack of notable associations, suggesting that different frequency bands operate relatively independently at anterior sites. **D-H.** Association patterns at all sites. **D.** 2D-association plots. **E-H.** Corresponding line plots showing the association strengths for major frequency bands as a function of *f*. The line plots are shown separately for the $\theta$-$\alpha$ (E), $\beta$ ($\beta_1$, $\beta_2$, $\beta_3$) (F), $\gamma_1$ (G), and $\gamma_2$ (H) bands to reduce clutter. The lower panels show representative 2D-association plots to highlight the specific bands included in the line plots. **E.** Line plots of association strengths for the $\theta$ and $\alpha$ frequencies, peaking at *Δf* ~ 3 Hz at posterior sites (especially in the regions shaded in gray). **F.** Line plots of association strengths for the $\beta$ ($\beta_1$, $\beta_2$, $\beta_3$) frequencies, peaking at *Δf* ~ 10 Hz at posterior sites (especially in the region shaded in gray). **G.** Line plots of association strengths for the $\gamma_1$ frequencies, peaking at *Δf* ~ 10 Hz at posterior sites (especially in the region shaded in gray). **H.** Line plots of association strengths for the $\gamma_2$ frequencies, peaking at *Δf* ~ 10 Hz at posterior sites and *Δf* ~ 16 Hz at lateral sites (especially in the regions shaded in gray). In all parts the shading on the line plots represents ±1 standard error of the mean with participants as the random effect.

across participants (the conventional *t*-value would be the ratio of the height of each peak to the width of the corresponding error region).

A representative lateral site shows associations for the $\theta$-$\alpha$ frequencies along the *Δf* ~ 3 Hz lines, for the $\beta_1$ frequencies along the *Δf* ~ 8 Hz lines, and particularly for the $\gamma_2$ frequencies along the *Δf* ~ 16 Hz lines (combined with negative associations along the *Δf* ~ 8 Hz lines) (e.g., the main panel in Fig 3B). The inter-participant consistency of these associations is reflected in the $\theta$ and $\alpha$ curves peaking at *Δf* ~ 3 Hz, the $\beta_1$ curve peaking at *Δf* ~ 8 Hz, and the $\gamma_2$ curve

substantially peaking at $\Delta f \sim$ 16 Hz and dipping at $\Delta f \sim$ 8 Hz (e.g., the upper-right panel in Fig 3B) relative to the corresponding error regions.

A representative anterior site shows the lack of any notable spectral-power associations (e.g., the main panel in Fig 3C) reflected in the relatively flat curves hovering around zero for all frequency bands (the upper-right panel in Fig 3C).

A second way in which we evaluated the consistency of cross-frequency association patterns was to confirm within-region similarity. In particular, the associations characterized by the $\theta$-$\alpha$ frequencies peaking at $\Delta f \sim$ 3 Hz (see the posterior region of Fig 3D and 3E) and the $\beta$-$\gamma$ frequencies peaking at $\Delta f \sim$ 10 Hz (see the posterior region of Fig 3D and 3F–3H) are shared among posterior sites, those characterized by the $\gamma_2$ frequencies peaking at $\Delta f \sim$ 16 Hz (see the lateral region of Fig 3D and 3H) are shared among lateral sites, and the relative lack of spectral-power associations (see the anterior region of Fig 3D–3H) are shared among anterior sites. A third way in which we evaluated data consistency was to confirm that the same conclusions about the spectral-power association patterns could be drawn based on the data from the odd or even numbered participants (see S1A–S1C Fig). We did not compute conventional $p$-values adjusted in various ways to more or less "control for" multiple comparisons because those would not add substantively to the above methods of evaluating reliability for the purpose of identifying characteristic patterns of spectral-power associations.

## Within-site (local) spectral-power associations on the (slower) seconds timescale

The within-site spectral-power associations on the seconds timescale (across 500 ms intervals) are characterized by vertical and horizontal directions. The representative association patterns (all involving positive associations) are shown in Fig 4A–4C. At a representative posterior site, large (top/bottom-15%) power variations in the $\theta$-low$\gamma$ frequencies (the $y$-axis indicating probe frequencies) were selectively associated with the power modulation in the $\alpha$ band and also mid$\beta$ band to a lesser degree (the $x$-axis indicating test frequencies), forming a vertical column at the $\alpha$ band and to a lesser degree at the mid$\beta$ band (e.g., the dashed rectangles in the lower panel in Fig 4A). The inter-participant consistency of these associations is reflected in the accompanying line plots in which the association strengths are averaged within each probe-frequency band (labeled with "$p$") and plotted as a function of test frequency. The $\alpha$-band column ($\alpha$-column) and the lesser mid$\beta$-band column are reflected in the $p\theta$, $p\alpha$, $p\beta_1$, $p\beta_2$, $p\beta_3$, and $p\gamma_1$ curves consistently peaking at the $\alpha$ band and also at mid$\beta$ band to a lesser degree (e.g., the upper panel in Fig 4A).

At a representative lateral site, power variations in the mid$\beta$-$\gamma$ frequencies were broadly associated with one another (e.g., the dashed square in the lower panel in Fig 4B), with the inter-participant consistency of these associations reflected in the $p\beta_2$, $p\beta_3$, $p\gamma_1$, and $p\gamma_2$ curves consistently elevated across the mid$\beta$-$\gamma$ range (e.g., the upper panel in Fig 4B).

At a representative anterior site, there were no notable spectral-power associations (e.g., the lower panel in Fig 4C), reflected in the curves representing all probe-frequency bands being relatively flat and hovering around zero (e.g., the upper panel in Fig 4C).

Fig 4D–4H shows that these characteristic patterns of spectral-power associations are consistent within each region. Specifically, the associations marked by the $\alpha$-column (and the mid$\beta$-column to a lesser degree) characterized by the $p\theta$, $p\alpha$, $p\beta_1$, $p\beta_2$, $p\beta_3$, and $p\gamma_1$ curves consistently peaking at the $\alpha$ band (and at the mid$\beta$ band to a lesser degree) are shared among posterior sites (see the posterior region of Fig 4D, 4E–4G). The associations marked by the broad square-shaped relationship across the mid$\beta$-$\gamma$ frequencies characterized by the $p\beta_2$, $p\beta_3$, $p\gamma_1$, and $p\gamma_2$ curves consistently elevated across the mid$\beta$-$\gamma$ range are shared among lateral sites

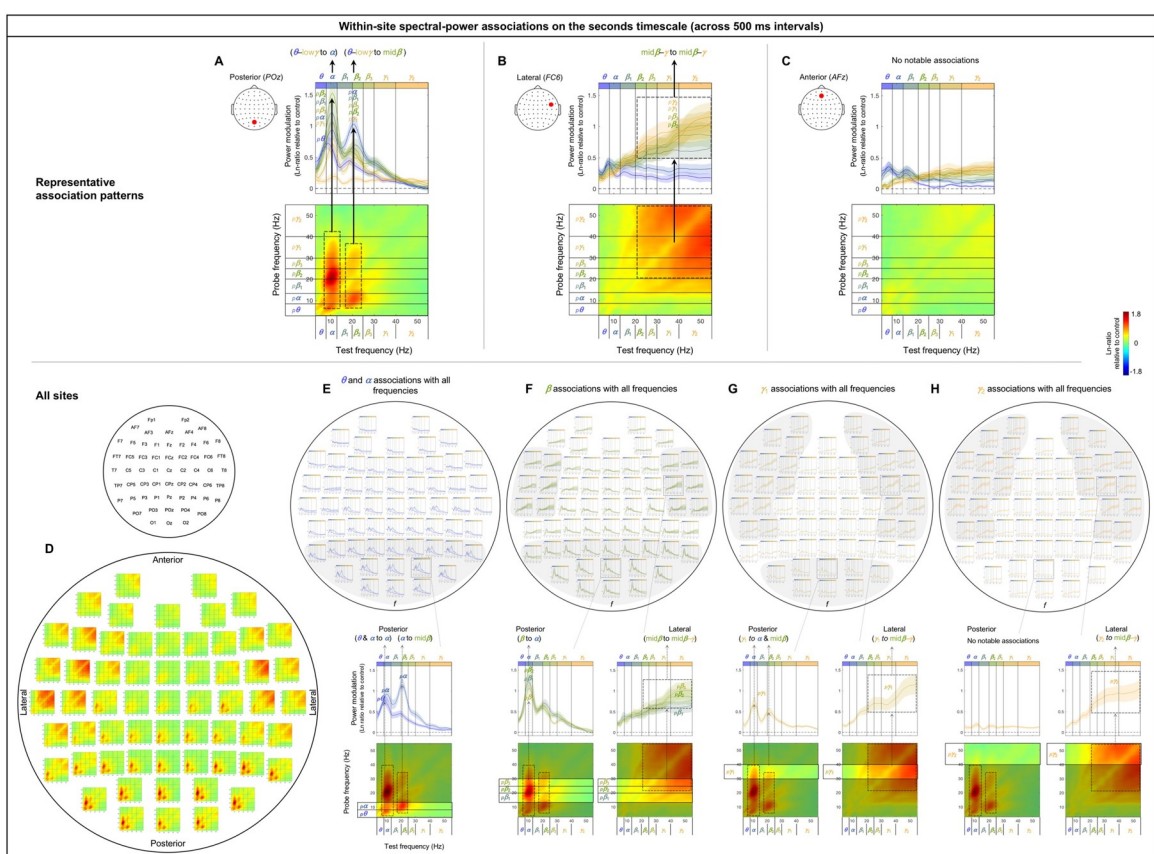

**Fig 4. Within-site spectral-power associations on the seconds timescale (across 500 ms intervals).** The seconds-timescale spectral-power associations were computed at 500 ms temporal resolution across the entire 5 min period (per site per participant) and averaged across participants. Each 2D-association plot shows the strengths of the same-signed (positive values, warmer colors) or opposite-signed (negative values, cooler colors) power modulation at all *test* frequencies (*x*-axis) that temporally coincided with the top/bottom-15% power variation at each *probe* frequency (*y*-axis), computed as Ln-ratio relative to the DCT-phase-scrambled control (to remove artifacts due to partial wavelet overlap; see text for details). Any asymmetry across the diagonal indicates directional differences; *X*-Hz power modulation coincident with the top/bottom-15% *Y*-Hz power variation may be stronger/weaker than the *Y*-Hz power modulation coincident with the top/bottom-15% *X*-Hz power variation. **A-C.** Representative association patterns. The strong positive associations along the vertical and horizontal directions indicate that spectral-power associations are characterized by multiple frequencies being associated with a specific frequency band or with one another. The accompanying line plots show the association strengths of each probe-frequency band (labeled with "*p*"; warmer-colored curves representing higher-probe-frequency bands) with all test frequencies (*x*-axis). The plots clearly show that the association strengths peaked at specific test frequencies. **A.** An example from a representative posterior site (*POz*) characterized by the associations of the **θ**-low**γ** frequencies with the **α** band and to a lesser degree with the mid**β** band (dashed rectangles), suggesting that the **α** band and to a lesser degree the mid**β** band play a role in coordinating cross-frequency interactions at posterior sites. **B.** An example from a representative lateral site (*FC6*) characterized by the broad associations among the mid**β**-**γ** frequencies (dashed square). **C.** An example from a representative anterior site (*AFz*) characterized by the lack of notable associations, suggesting that different frequency bands operate relatively independently at anterior sites. **D-H.** Association patterns at all sites. **D.** 2D-association plots. **E-H.** Corresponding line plots showing the association strengths of each probe-frequency band with all test frequencies (*x*-axis). The line plots are shown separately for the **θ-α** (E), **β** (**β₁**, **β₂**, **β₃**) (F), **γ₁** (G), and **γ₂** (H) bands to reduce clutter. The lower panels show representative 2D-association plots to highlight the specific probe-frequency bands included in the line plots. **E.** Line plots showing the association strengths of the **θ** and **α** probe-frequency bands (labeled "*p*θ" and "*p*α") with all test frequencies. The **θ** and **α** bands are both associated with the **α** band while the **α** band is additionally associated with the mid**β** band at posterior sites (especially in the region shaded in gray). **F.** Line plots showing the association strengths of the **β** (**β₁**, **β₂**, **β₃**) probe-frequency bands (labeled "*p*β₁," "*p*β₃," and "*p*β₃") with all test frequencies. The **β** (**β₁**, **β₂**, **β₃**) bands are primarily associated with the **α** band at posterior sites, while the **β₂** and **β₃** bands are broadly associated with the mid**β**-**γ** bands at lateral sites (especially in the regions shaded in gray). **G.** Line plots showing the association strengths of the **γ₁** probe-frequency band (labeled "*p*γ₁") with all test frequencies. The **γ₁** band is associated with the **α** and mid**β** bands at posterior sites, while it is broadly associated with the mid**β**-**γ** bands at lateral sites (especially in the regions shaded in gray). **H.** Line plots showing the association strengths of the **γ₂** probe-band (labeled "*p*γ₂") with all test-frequency bands. The **γ₂** band is broadly associated with the mid**β**-**γ** bands at lateral sites (especially in the regions shaded in gray). In all parts the shading on the line plots represents ±1 standard error of the mean with participants as the random effect.

(see the lateral regions of Fig 4D, 4F–4H). The lack of notable spectral-power associations characterized by the curves for the entire range of probe frequencies being relatively flat and hovering around zero is shared among anterior sites (see the anterior region of Fig 4D, 4E–4H). Further, these characteristic patterns of spectral-power associations are evident in the data from both the odd and even numbered participants (S1D–S1F Fig).

In summary, the patterns of within-site (local) spectral-power associations (nearly exclusively positive) were distinct for the sub-second (within a 500 ms interval) and seconds (across 500 ms intervals) timescales. The (fast) sub-second-timescale associations were characterized by 45˚ lines parallel to the diagonal, indicative of associations at constant $f$'s, whereas the (slower) seconds-timescale associations were characterized by columns, indicative of multiple frequencies being selectively associated with a specific band or with one another. In particular, the (fast) sub-second-timescale oblique associations were primarily characterized by the $\theta$-$\alpha$ frequencies associated at $\Delta f \sim$ 3 Hz and the $\beta$-$\gamma$ frequencies associated at $\Delta f \sim$ 10 Hz at posterior sites (e.g., Fig 3A; also see the posterior region of Fig 3D, 3E–3H), and the $\beta$-$\gamma$ frequencies associated at $\Delta f \sim$ 16 Hz at lateral sites (e.g., Fig 3B; also see Fig the lateral regions of 3D, 3H). The (slower) seconds-timescale columnar associations were primarily characterized by the $\theta$-low$\gamma$ frequencies associated with the $\alpha$ band (also mid$\beta$ band to a lesser degree) at posterior sites (e.g., Fig 4A; also see the posterior region of Fig 4D, 4E–4G), and the mid$\beta$-$\gamma$ frequencies broadly associated with one another at lateral sites (e.g., Fig 4B; also see the lateral regions of Fig 4D, 4F–4H). No notable spectral-power associations were found on either timescale at anterior sites (e.g., Figs 3C and 4C; also see the anterior region of Figs 3D, 3E–3H, 4D and 4E–4H).

## Cross-site (long-distance) spectral-power associations on the (fast) sub-second timescale

The cross-site spectral-power associations (in Ln ratio) on the sub-second-timescale (within a 500 ms interval) are characterized by the diagonal, indicative of within-frequency interactions, while the ranges of associated frequencies varied from site-pair to site-pair (representative examples shown in Fig 5). While there are 60 x 59 (3,540) site combinations, characteristic patterns of cross-site associations can be identified by examining the associations from all sites to representative sites: *Oz* for the posterior region, *Cz* for the central region, *AFz* for the anterior region, and *T7* and *T8* for the left-lateral and right-lateral regions, respectively. Examining cross-site associations in this way (i.e., from all sites to a target site) allowed us to directly compare the impact of each site on a target site by evaluating the spectral-power modulation (in Ln-ratio) at the target site coincident with large (top/bottom-15%) spectral-power variations at each of the other sites.

The spectral-power associations to *Oz* (a representative posterior site) from all other sites are shown in Fig 6A. The inter-participant consistency of the association patterns was evaluated in three ways. Because all associations were characteristically diagonal (Fig 6A), we computed the diagonal magnitudes (Fig 6B) with the gray regions indicating ±1 standard error of the mean (with participants as the random effects). Large frequency-dependent variations relative to the widths of the error regions indicate statistical reliability (the conventional *t*-values being their ratio). A second way in which we evaluated data consistency was to note bilateral symmetry. While large-scale neural interactions are expected to show general bilateral symmetry, spurious patterns would not. A third way was to confirm that the same conclusions about the association patterns could be drawn based on the data from the odd or even numbered participants (see S2 and S3 Figs).

Although we applied the surface-Laplacian transform to all EEG data to reduce the influences of volume conduction, diagonal associations could reflect volume conduction especially

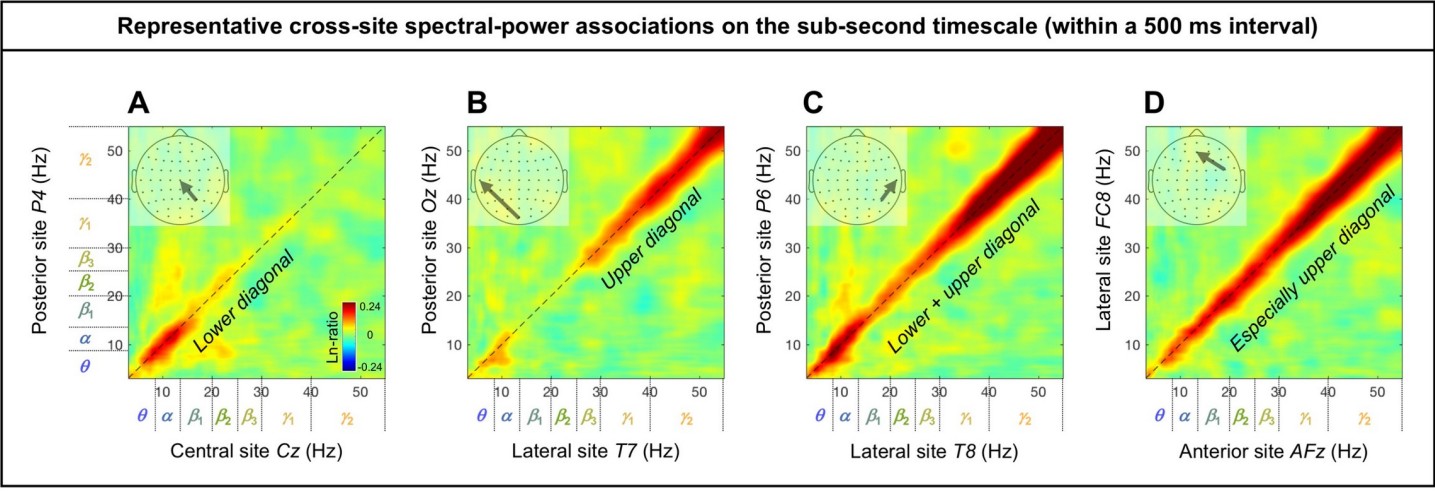

**Fig 5. Representative examples of cross-site spectral-power associations on the sub-second timescale (within a 500 ms interval).** Each 2D spectral-power-association plot shows the strengths of the same-signed (positive values, warmer colors) or opposite-signed (negative values, cooler colors) power modulation (in Ln-ratio) for all test-site frequencies (*x*-axis) that temporally coincided with the top/bottom-15% power variation at each probe-site frequency (*y*-axis). The sub-second-timescale spectral-power associations were computed at 512 Hz temporal resolution within each 500 ms interval, averaged across all intervals and then across participants. The associations (all positive) were characteristically diagonal, indicative of within-frequency associations, while the ranges of associated frequencies varied from site-pair to site-pair. **A.** Within-frequency associations in the lower diagonal involving the **θ**-low**β** frequencies from a posterior site *P4* to a central site *Cz*. **B.** Within-frequency associations in the upper diagonal involving the *γ* frequencies from a posterior site *Oz* to a lateral site *T7*. **C.** Within-frequency associations in the lower and upper diagonal involving the *θ*-*α* and *γ* frequencies from a posterior site *P6* to a lateral site *T8*. **D.** Stronger within-frequency associations in the higher frequencies from a lateral site *FC8* to an anterior site *AFz*.

between adjacent sites (e.g., [32]). Nevertheless, any spurious contributions from volume conduction would be largely frequency independent because spectral-power profiles tended to be similar between adjacent sites. For example, although the overall strong diagonal associations to *Oz* from its lateral neighbors, *O1* and *O2*, may primarily reflect volume conduction (Fig 6A and 6B), the fact that the associations were relatively weak in the *α* band (Fig 6B) may reveal a characteristic of cross-site neural interactions.

Overall, the sub-second-timescale cross-site spectral-power associations to *Oz* (a representative posterior site) from other sites were all diagonal (within-frequency) and generally diminished with distance (Fig 6A and 6B), except that the *γ*-band associations from lateral sites (especially from *T7* [Fig 6C] and *T8*) remained strong. In general, the within-frequency associations to *Oz* were dominated by the *α* or *θ*-*α* bands (e.g., Fig 6D and 6F) or by combinations of the *θ*-*α* and *γ* bands (e.g., Fig 6C and 6E), but were never dominated by the *β* band (also see Fig 6A and 6B).

The sub-second-timescale cross-site spectral-power associations to *Cz* (a representative central site) from other sites were also all diagonal (within-frequency) and generally diminished with distance (Fig 7A and 7B), except that the *α*-band associations from posterior sites remained strong (e.g., Fig 7D; also see the posterior region of Fig 7A and 7B). The within-frequency associations to *Cz* were generally dominated by the *α* band (e.g., Fig 7C–7F), while some associations from within the central region were relatively frequency independent (e.g., from *FC3*, *FC4*, *C3*, *C4*; also see Fig 7A and 7B).

The sub-second-timescale cross-site spectral-power associations to *AFz* (a representative anterior site) from other sites were also all diagonal (within-frequency) and generally diminished with distance (Fig 8A and 8B), except that the *α*-band associations from some posterior sites remained strong (e.g., from *P7*, *PO7*, *P8* [Fig 8F], *PO8*). The within-frequency associations to *AFz* were generally dominated by the *α* band (e.g., Fig 8C and 8F), the *β*-*γ* bands (e.g.,

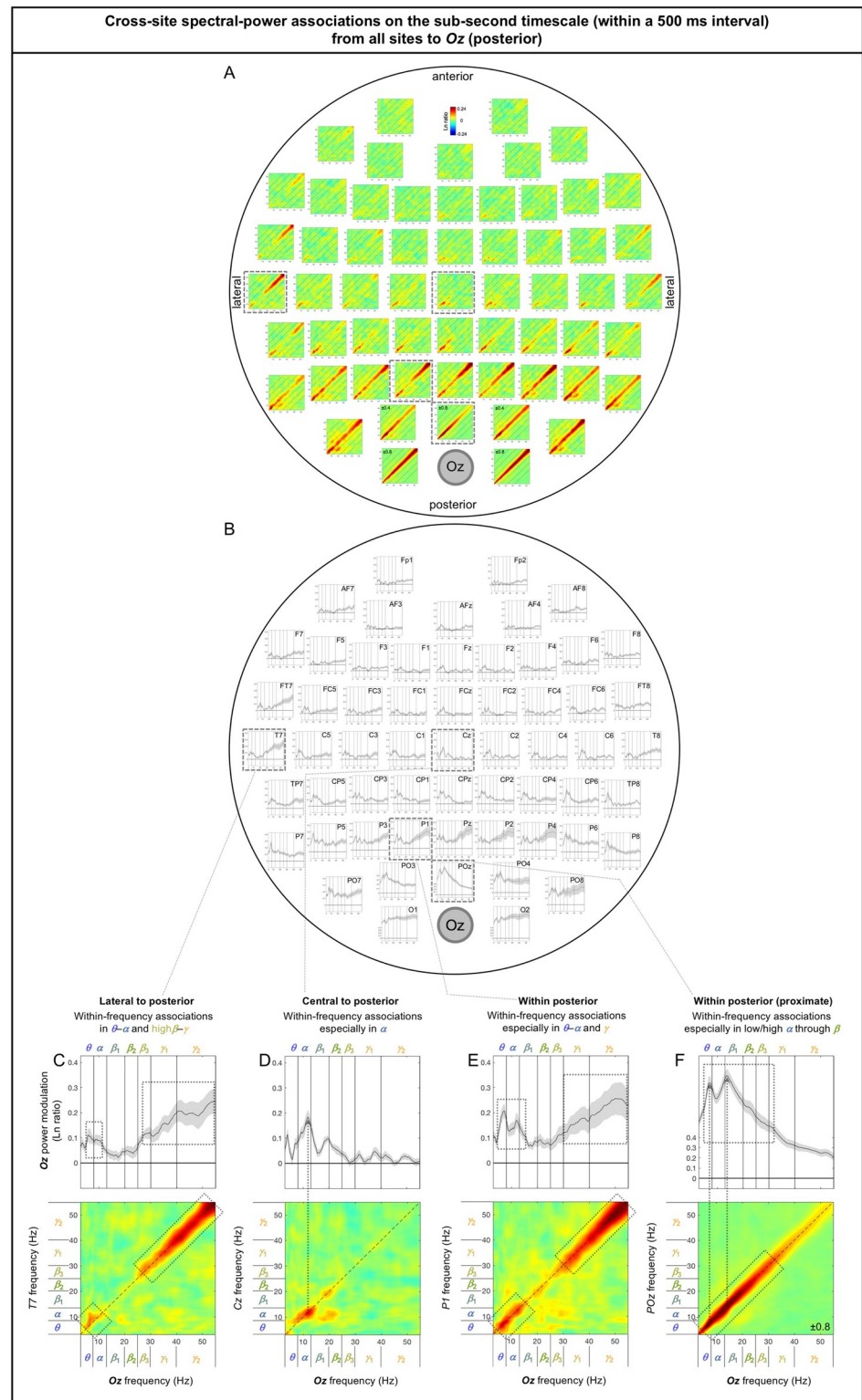

**Fig 6. Cross-site spectral-power associations on the sub-second timescale (within a 500 ms interval) from all sites to *Oz* (a representative posterior site). A.** 2D spectral-power association plots. All associations were characteristically diagonal, indicative of within-frequency associations, while the ranges of associated frequencies varied across site-pairs. **B.** The values along the diagonal with the shaded regions indicating ±1 standard error of the mean (with participants as the random effect). **C-F.** Representative associations. **C.** *T7* (a left-lateral site) to *Oz* spectral-power

associations, characterized by within-frequency associations in the *θ-α* and high*β-γ* bands, skipping the middle (*β* band). **D.** *Cz* (a central site) to *Oz* spectral-power associations, characterized by within-frequency associations primarily in the *α* band. **E.** *P1* (a posterior site) to *Oz* spectral-power associations, characterized by within-frequency associations particularly in the *θ-α* and *γ* bands, skipping the middle (*β*) band. **F.** *POz* (an adjacent site) to *Oz* spectral-power associations, characterized by within-frequency associations particularly in the *θ* through low*β* bands. The color scale is shown at the top of part A (also indicated within association plots when different; for example, "±0.4" indicates that the color scale for the corresponding association plot ranges from –0.4 to +0.4).

Fig 8D), or combinations of the *α* and *β-γ* bands (e.g., Fig 8E), but were never dominated by the *β* band (also see Fig 8A and 8B). It is noteworthy that the associations to *AFz* from a pair of neighboring sites, *Fp1* (Fig 8C) and *AF7* (Fig 8D) were opposite, that is, the associations from *Fp1* were dominated by the *α* band whereas those from *AF7* were dominated by the *β-γ* bands; this spectrally opposite relationship was replicated on the right side for the associations from *Fp2* and *AF8* (Fig 8A and 8B) and obtained from both the odd and even numbered participants (S2G–S2I Fig). These spectrally opposite association patterns from neighboring sites indirectly confirm that the surface-Laplacian transform was reasonably effective at revealing distinct neural sources.

The sub-second-timescale cross-site spectral-power associations to *T7* and *T8* (representative left and right lateral sites) from other sites were also all diagonal (within-frequency) and generally diminished with distance (Fig 9A and 9B, 9H and 9I), except that the *α*-band associations from some contralateral sites (e.g., *C6* [Fig 9G] and *P8* for *T7* and *C5* [Fig 9J] and *P7* for *T8*) and the high*β-γ* associations from *Oz* (Fig 9F and 9K) remained strong. The within-frequency associations to *T7* and *T8* were generally dominated by the *β-γ*, high*β-γ*, or *γ* band (primarily from the ipsilateral central and anterior regions; e.g., Fig 9C and 9D, 9M and 9N), the *α* band (primarily from the contralateral regions; e.g., Fig 9G and 9J), or combinations of the *θ-α* or *α* band and the high*β-γ* or *γ* band (primarily from ipsilateral posterior and central regions; Fig 9E and 9F, 9K and 9L), but were never dominated by the *β* band (also see Fig 14A and 14B, 14H and 14I).

Note that the sub-second-timescale cross-site association patterns to *Oz* (posterior), *Cz* (central), *AFz* (anterior), *T7* (left lateral), and *T8* (right lateral) were generally bilaterally symmetric (Fig 6A and 6B [posterior], 7A-B [central], 8A-B [anterior], 9A vs. 9H, 9B vs. 9I, 9C-G vs. 9J-N [lateral]). Further, the above conclusions hold for data from either the odd or even numbered participants (see S2A–S2C Fig [posterior], S2D–S2F Fig [central], S2G–S2I Fig [anterior], S2J–S2L Fig, S2M–S2O Fig [lateral]).

In summary, the (fast) sub-second-timescale (within a 500 ms interval) cross-site (long-distance) spectral-power associations (all positive) are characterized by diagonal, within-frequency associations dominated by different frequency ranges. Although the frequency dependence varied from site-pair to site-pair, some characteristic patterns emerged. Not surprisingly, the association strengths generally diminished with distance, though some associations remained strong. Those "super" long-distance associations tended to be dominated by the *α* band, such as the associations from inferior posterior sites (*PO7*, *PO3*, *POz*, *PO4*, *PO8*, *O1*, *Oz*, and *O2*) to a representative central site, *Cz* (Fig 7A and 7B), from lateral posterior sites (*P7*, *P5*, *PO7*, *P6*, *P8*, and *PO8*) to a representative anterior site, *AFz* (Fig 8A and 8B), and from contralateral sites (*FT8*, *C6*, *TP8*, *P8*, and *PO8*; *FT7*, *C5*, *TP7*, *P8*, and *PO7*) to representative left and right lateral sites *T7* and *T8* (Fig 9A and 9B, 9H and 9I). A few super long-distance associations were dominated by the *β-γ* bands, such as those from a posterior site, *Oz*, to representative left and right lateral sites, *T7* and *T8* (Fig 9A–9B, 9H–9I). Overall, the associations to *Oz* (a representative posterior site), *AFz* (a representative anterior site), and *T7/T8* (representative left and right lateral sites) were dominated by the *θ-α* bands, *β-γ* bands, or combinations

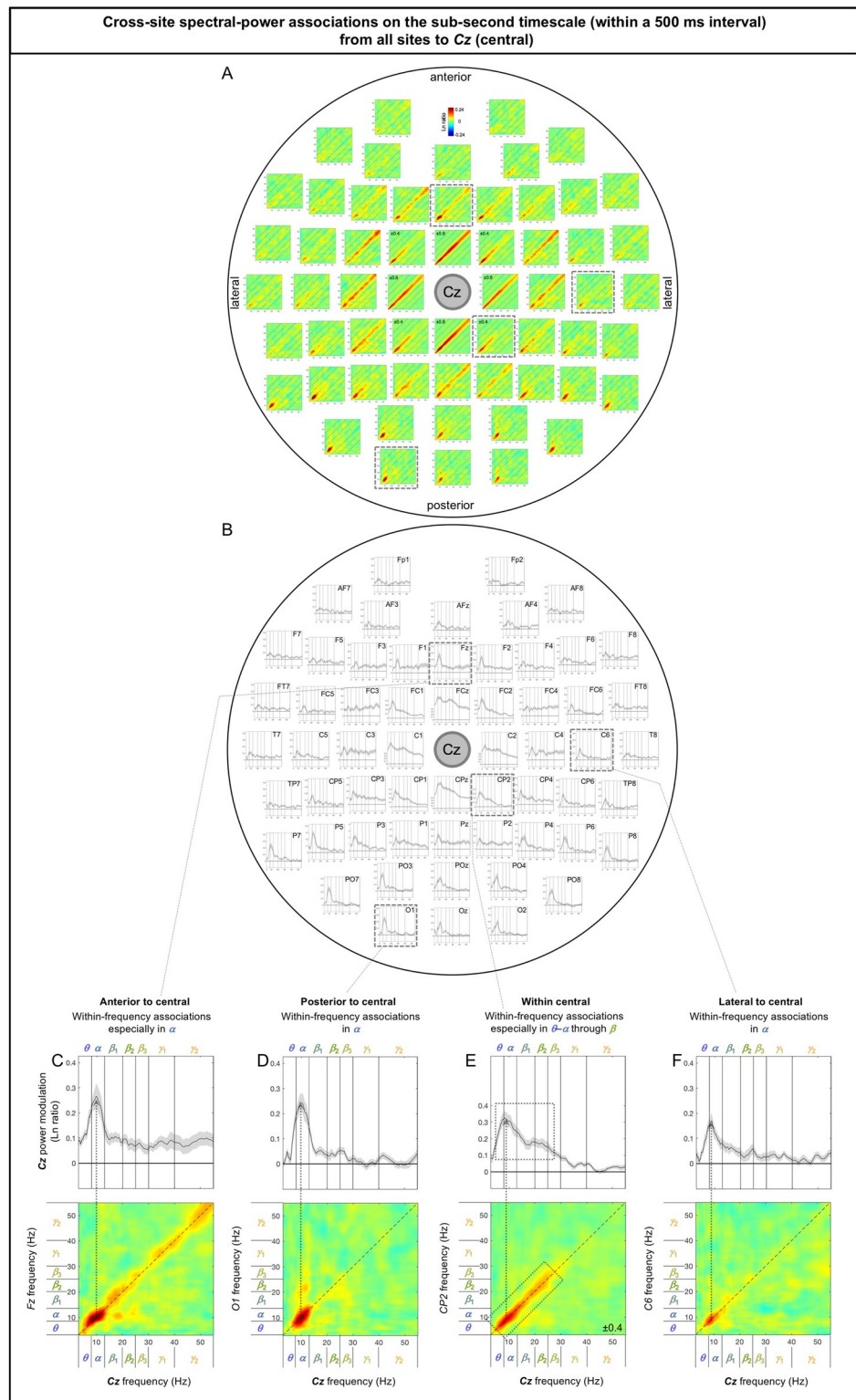

**Fig 7. Cross-site spectral-power associations on the sub-second timescale (within a 500 ms interval) from all sites to *Cz* (a representative central site). A.** 2D spectral-power association plots. All associations were characteristically diagonal, indicative of within-frequency associations, while the ranges of associated frequencies varied across site-pairs. **B.** The values along the diagonal with the shaded regions indicating ±1 standard error of the mean (with participants as the random effect). **C-F.** Representative associations. **C.** *Fz* (an anterior site) to *Cz* spectral-power

associations, characterized by within-frequency associations particularly in the **α** band. **D.** *O1* (a posterior site) to *Cz* spectral-power associations, characterized by within-frequency associations in the **α** band. **E.** *CP2* (a central site) to *Cz* spectral-power associations, characterized by within-frequency associations particularly in the **θ-α** through **β** bands. **F.** *C6* (a lateral site) to *Cz* spectral-power associations, characterized by within-frequency associations in the **α** band. The color scale is shown at the top of part A (also indicated within association plots when different; for example, "±0.4" indicates that the color scale for the corresponding association plot ranges from –0.4 to +0.4).

of the $\theta$-$\alpha$ and $\beta$-$\gamma$ bands, whereas the associations to *Cz* (a representative central site) were dominated by the $\alpha$ band. Notably, no associations were dominated by the $\beta$ band.

## Cross-site (long-distance) spectral-power associations on the (slower) seconds timescale

The cross-site spectral-power associations (in Ln-ratio) on the seconds timescale (across 500 ms intervals) are characterized by columnar, square, and diagonal patterns. Notably, the associations from posterior sites to other sites are characterized by "$\alpha$-columns," that is, the posterior spectral-power variations in the $\theta$-low$\gamma$ bands were selectively associated with the $\alpha$-band power modulation at other sites (e.g., Fig 10A–10D). Other associations are characterized by the $\alpha$-band to $\alpha$-band associations (e.g., Fig 10E–10H), broad associations among the $\beta$-$\gamma$ bands (Fig 10F and 10H to some degree), and within-frequency (diagonal) associations (e.g., Fig 10H).

As we did above for the sub-second-timescale associations, we examined the associations from all sites to representative sites in the posterior (*Oz*), central (*Cz*), anterior (*AFz*), left-lateral (*T7*), and right-lateral (*T8*) regions. To assess the inter-participant consistency of the association patterns, we generated line plots corresponding to each 2D-association plot, where each curve shows the test-site spectral-power modulation coincident with the probe-site top/bottom-15% power variation in each major frequency band (lower-to-higher frequency bands labeled with cooler-to-warmer colors), with the shaded regions indicating ±1 standard error of the mean (with participants as the random effect). In other words, each curve shows a horizontal slice of the corresponding 2D-association plot for a specific probe-site frequency band (see the rectangular boxes in the bottom panels in Figs 11–14). Overall, spectral-power associations to all five target sites diminished with distance (Figs 11–14) as they did on the sub-second timescale.

The seconds-timescale cross-site spectral-power associations to *Oz* (a representative posterior site) from other sites were characterized by two general patterns depending on the region. The associations from within the posterior region to *Oz* were characterized by (1) the $\alpha$-low$\gamma$-band (posterior) to $\alpha$-band (*Oz*) associations, $\alpha$-columns, and (2) the $\alpha$-&-mid$\beta$-band (posterior) to mid$\beta$-band (*Oz*) associations to a lesser degree (e.g., Fig 11C–11D; also see the posterior region of Fig 11A–11B). The associations from outside the posterior region to *Oz* were characterized by the $\alpha$-band (non-posterior) to $\alpha$-band (and to mid$\beta$-band to a lesser degree) (*Oz*) associations (e.g., Fig 11E–11F; also see the inferior-anterior, central, and lateral regions of Fig 11A–11B). In addition, the associations from proximate sites to *Oz* included within-frequency (diagonal) associations such as the $\theta$-$\beta$-band associations from *POz* (Fig 11D) and broadband associations from *O1* and *O2* (see the bottom of Fig 11A), though the latter may primarily reflect volume conduction.

The seconds-timescale cross-site spectral-power associations to *Cz* (a representative central site) from posterior sites were characterized by the $\theta$-low$\gamma$-band (posterior) to $\alpha$-band (*Cz*) associations, $\alpha$-columns (e.g., Fig 12F; also see the posterior region of Fig 12A–12B). The associations from non-posterior sites to *Cz* generally included the $\alpha$-band (non-posterior) to $\alpha$-band (*Cz*) associations (e.g., Fig 12C–12E; also see the anterior, central, and lateral regions of Fig 12A–12B). In addition, the associations from proximate sites to *Cz* included within-

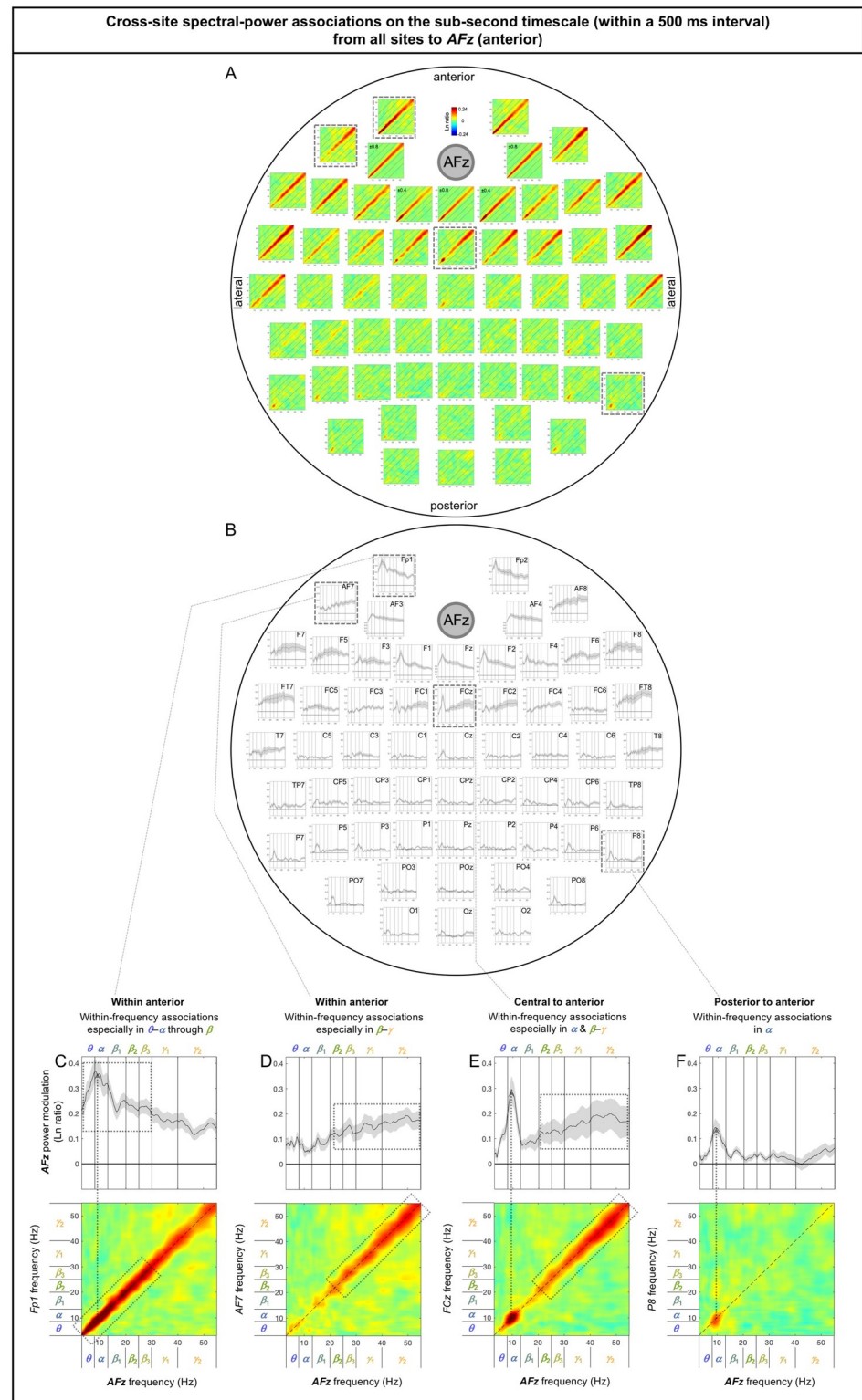

**Fig 8. Cross-site spectral-power associations on the sub-second timescale (within a 500 ms interval) from all sites to *AFz* (a representative anterior site). A.** 2D spectral-power association plots. All associations were characteristically diagonal, indicative of within-frequency associations, while the ranges of associated frequencies varied across site-pairs. **B.** The values along the diagonal with the shaded regions indicating ±1 standard error of the mean (with participants as the random effect). **C-F.** Representative associations. **C.** *Fp1* (an anterior site) to *AFz* spectral-power

associations, characterized by within-frequency associations particularly in the $\theta$-$\alpha$ through $\beta$ bands. **D.** *AF7* (an anterior site) to *AFz* spectral-power associations, characterized by within-frequency associations particularly in the $\beta$-$\gamma$ bands. Note that although *Fp1* and *AF7* are adjacent anterior sites the frequency characteristics of their associations with *AFz* were opposite (the low-to-middle bands for *Fp1* and the middle-to-high bands for *AF7*); this pattern was replicated on the right side at *Fp2* and *AF8* (see A and B). **E.** *FCz* (a central site) to *AFz* spectral-power associations, characterized by within-frequency associations particularly in the $\alpha$ and mid$\beta$-$\gamma$ bands. **F.** *P6* (a posterior site) to *AFz* spectral-power associations, characterized by within-frequency associations in the $\alpha$ band. The color scale is shown at the top of part A (also indicated within association plots when different; for example, "±0.4" indicates that the color scale for the corresponding association plot ranges from −0.4 to +0.4).

frequency (diagonal) associations in the $\theta$-$\beta$ bands (e.g., Fig 12E; also see the sites surrounding *Cz* in Fig 12A–12B).

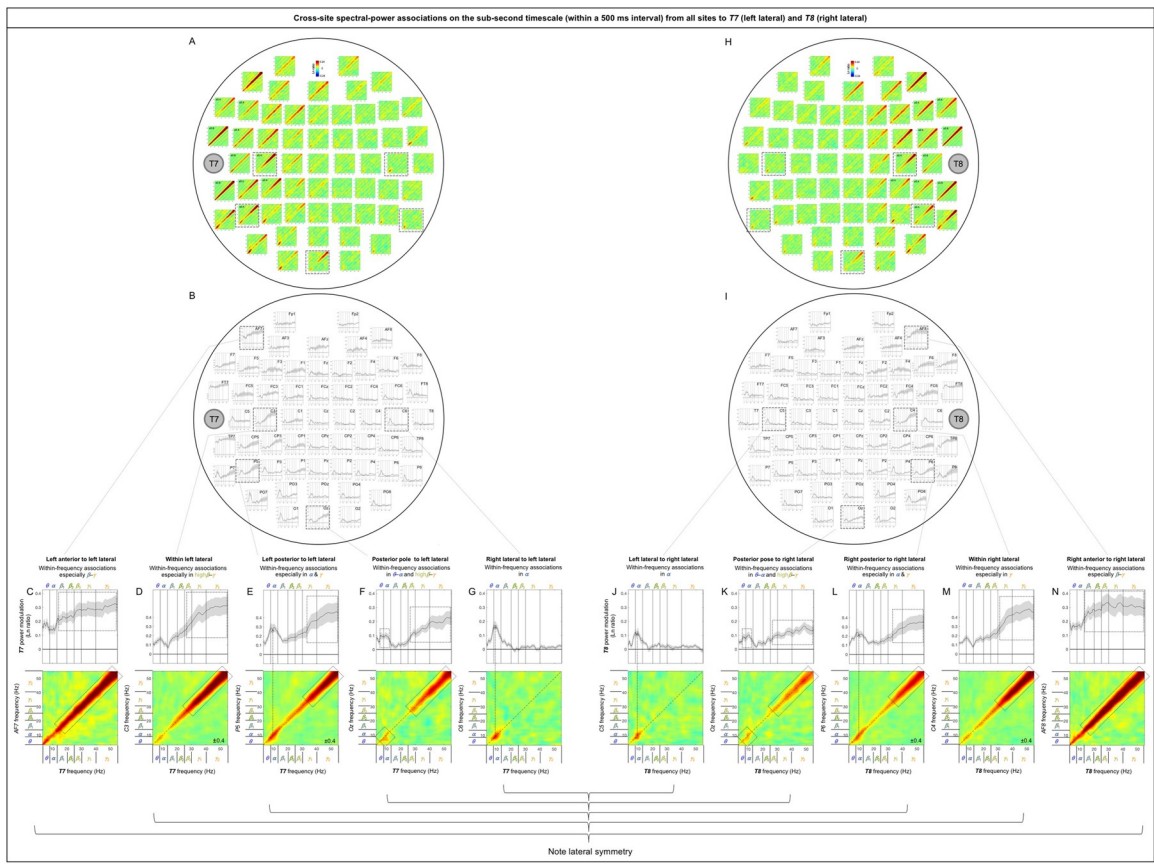

**Fig 9. Cross-site spectral-power associations on the sub-second time scale (within a 500 ms interval) from all sites to *T7* (a representative left lateral site) and *T8* (a representative right lateral site). A, H.** 2D spectral-power association plots for *T7* and *T8*. All associations were characteristically diagonal, indicative of within-frequency associations, while the ranges of associated frequencies varied across site-pairs. **B, I.** The power-modulation values along the diagonal with the shaded regions indicating ±1 standard error of the mean (with participants as the random effect). **C-G, J-N.** Representative associations for *T7* or *T8*. Note the bilateral symmetry. **C, N.** *AF7/AF8* (ipsilateral anterior sites) to *T7/T8* spectral-power associations, characterized by within-frequency associations particularly in the $\beta$-$\gamma$ bands. **D, M.** *C3/C4* (ipsilateral sites) to *T7/T8* spectral-power associations, characterized by within-frequency associations in the high$\beta$-$\gamma$ bands (D) or the $\gamma$ band (M). **E, L.** *P5/P6* (ipsilateral posterior sites) to *T7/T8* spectral-power associations, characterized by within-frequency associations particularly in the $\alpha$ and $\gamma$ bands. **F, K.** *Oz* (a posterior site) to *T7/T8* spectral-power associations, characterized by within-frequency associations in the $\theta$-$\alpha$ and high$\beta$-$\gamma$ bands. **G, J.** *C6/C5* (contralateral sites) to *T7/T8* spectral-power associations, characterized by within-frequency associations in the $\alpha$ band. The color scale is shown at the top of part A (also indicated within association plots when different; for example, "±0.4" indicates that the color scale for the corresponding association plot ranges from −0.4 to +0.4).

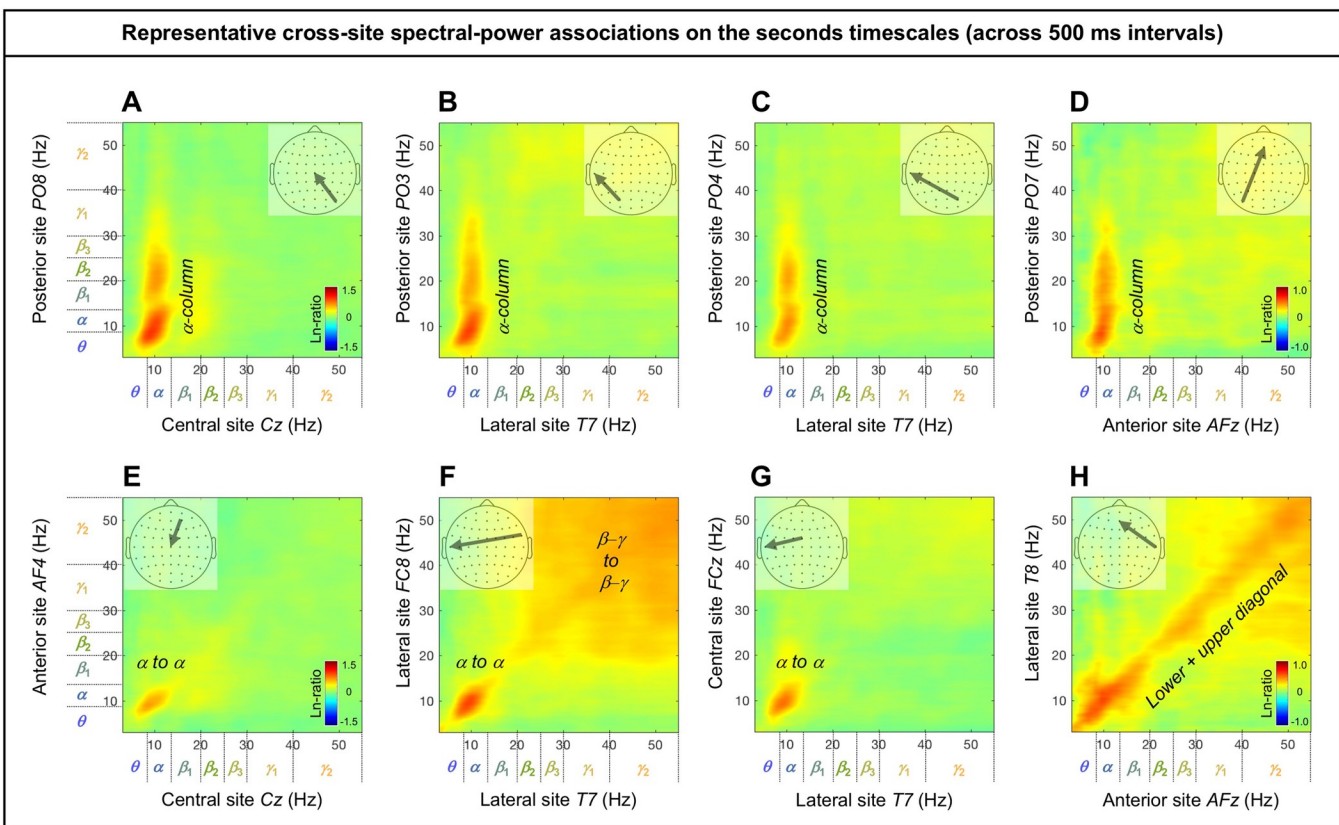

**Fig 10. Representative examples of cross-site spectral-power associations on the seconds timescale (across 500 ms intervals).** Each 2D spectral-power-association plot shows the strengths of the same-signed (positive values, warmer colors) or opposite-signed (negative values, cooler colors) power modulation (in Ln-ratio) for all test-site frequencies (*x*-axis) that temporally coincided with the top/bottom-15% power variation at each probe-site frequency (*y*-axis). The seconds-timescale spectral-power associations were computed at 500 ms temporal resolution across the entire 5 min period, and averaged across participants. We obtained several characteristic patterns of associations. **A-D.** The "$\alpha$-column" associations from posterior sites to other sites characterized by the posterior $\theta$-low$\gamma$ bands selectively associated with the **$\alpha$** band at other sites. **E-H.** Other patterns of associations (not involving posterior sites) characterized by the **$\alpha$** to **$\alpha$** associations (**E-H**), broad associations among the **$\beta$-$\gamma$** bands (**F** and **H** to some degree), and within-frequency (diagonal) associations (**H**).

The seconds-timescale cross-site spectral-power associations to *AFz* (a representative anterior site) from posterior sites were characterized by the $\theta$-low$\gamma$-band (posterior) to $\alpha$-band (*AFz*) associations, $\alpha$-columns (e.g., Fig 13F; also see the posterior region of Fig 13A–13B). The associations from non-posterior sites generally included the $\alpha$-band (non-posterior) to $\alpha$-band (*AFz*) associations (e.g., Fig 13E; also see the anterior, central, and lateral regions of Fig 13A–13B). In addition, the associations from anterior and lateral sites to *AFz* included broad associations in the high$\beta$-$\gamma$ band as well as within-frequency (diagonal) associations (e.g., Fig 13C–13D; also see the anterior and dorsal-lateral regions of Fig 13A–13B).

The seconds-timescale cross-site spectral-power associations to *T7* and *T8* (representative left and right lateral sites) from posterior sites were characterized by the $\theta$-low$\gamma$-band (posterior) to $\alpha$-band (*T7/T8*) associations, $\alpha$-columns (e.g., Fig 14E, 14L; also see the posterior region of Fig 14A–14B, 14H–14I). The associations from non-posterior sites to *T7* and *T8* generally included the $\alpha$-band (non-posterior) to $\alpha$-band (*T7/T8*) associations (e.g., Fig 14C–14D, 14F–14G, 14J–14K, 14M–14N; also see the anterior, central, and lateral regions of Fig 14A–14B, 14H–14I). In addition, the associations from the ipsilateral and contralateral sites to *T7* and *T8* included broad associations in the high$\beta$-$\gamma$ bands (e.g., Fig 14C–14D, 14G, 14J, 14M–14N; also see the lateral regions of Fig 14A–14B, 14H–14I) as well as within-frequency (diagonal) associations from the ipsilateral sites (e.g., Fig 14C–14D, 14M–14N).

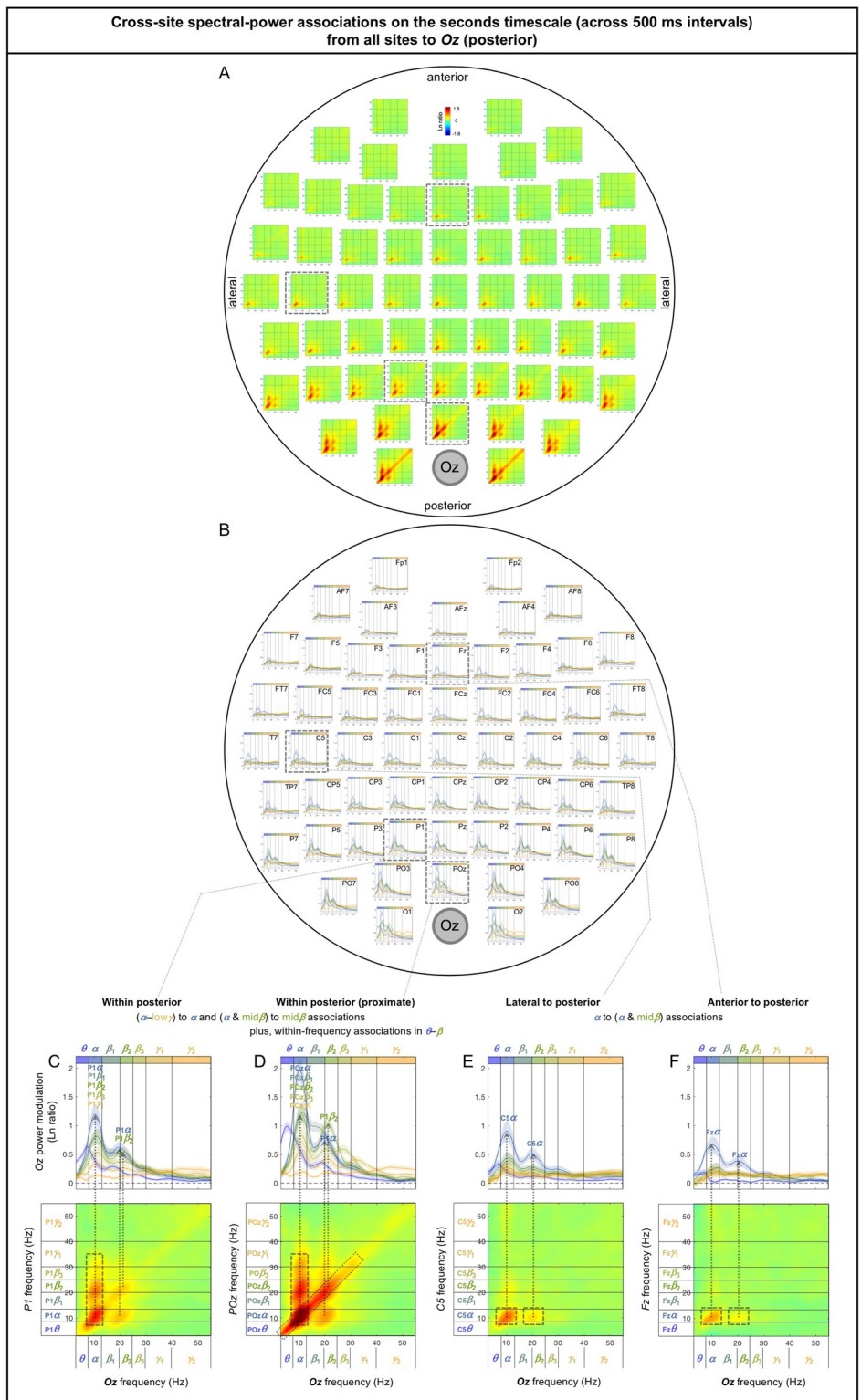

**Fig 11. Cross-site spectral-power associations on the seconds timescale (across 500 ms intervals) from all sites to *Oz* (a representative posterior site). A.** 2D spectral-power association plots. Some associations are columnar, indicating multiple frequencies associated with a specific (primarily *α*) band, and some are diagonal, indicating within-frequency associations. **B.** Replotting of A. Each curve shows the spectral-power modulation at *Oz* associated with probe-site power variation in each major frequency band (lower-to-higher frequency bands labeled with cooler-to-

warmer colors), with the shaded regions indicating ±1 standard error of the mean (with participants as the random effect). In other words, each curve shows a horizontal slice of a 2D-association plot for a specific probe-site frequency band (see the rectangular boxes in the bottom panels). **C-F.** Representative associations. **C.** *P1* (a posterior site) to *Oz* spectral-power associations, characterized by the $\alpha$-low$\gamma$ bands at *P1* associated with the $\alpha$-band at *Oz* ($\alpha$-column), and the $\alpha$-&-mid$\beta$ bands at *P1* somewhat associated the mid$\beta$ band at *Oz*. **D.** *POz* (an adjacent site) to *Oz* spectral-power associations, characterized by the $\alpha$-low$\gamma$ bands at *POz* associated with the $\alpha$ band at *Oz* ($\alpha$-column), the $\alpha$-$\beta$ bands at *POz* somewhat associated with the mid$\beta$ band at *Oz* (as in C, but stronger), and the diagonal (within-frequency) associations in the $\theta$-$\beta$ bands. **E.** *C5* (a lateral site) to *Oz* spectral-power associations, characterized by the $\alpha$ band at *C5* associated with the $\alpha$ band (and the mid$\beta$-band to some degree) at *Oz*. **F.** *Fz* (an anterior site) to *Oz* spectral-power associations, characterized by the $\alpha$ band at *Fz* associated with the $\alpha$ band (and the mid$\beta$ band to some degree) at *Oz* (as in E). The color scale is shown at the top of part A. Note that spectral-power associations from most posterior sites to *Oz* contain $\alpha$-columns. The color scale is shown at the top of part A.

Note that the seconds-timescale cross-site association patterns to *Oz* (posterior), *Cz* (central), *AFz* (anterior), *T7* (left lateral), and *T8* (right lateral) were generally bilaterally symmetric (Fig 11A–11B [posterior], Fig 12A–12B [central], Fig 13A–13B [anterior], Fig 14A vs. 14H, Fig 14B vs. 14I, Fig 14C–14G vs. 14J–14N [lateral]). Further, the above conclusions hold for data from either the odd or even numbered participants (see S3A–S3C Fig [posterior], S3D–S3F Fig [central], S3G–S3I Fig [anterior], S3J–S3L Fig, S3M–S3O Fig [lateral]).

In summary, the (slower) seconds-timescale (across 500 ms intervals) cross-site (long-distance) spectral-power associations (all positive) are characterized by the $\alpha$-band to $\alpha$-band associations prevalent across most site pairs (Figs 11–14), within-frequency (diagonal) associations from proximate sites (Figs 11–14), broad high$\beta$-$\gamma$-band associations involving lateral and anterior targets (Figs 13 and 14), and notably by the $\theta$-low$\gamma$-band to $\alpha$-band associations, $\alpha$-columns, from posterior sites to other sites (Figs 12–14).

## General discussion

The observed EEG spectral power intrinsically fluctuated on at least two distinct timescales, the sub-second timescale (within a 500 ms interval) with the low and high states typically lasting ~230 ms, and the seconds timescale (across 500 ms intervals) with the low and high states typically lasting ~3.75 s (Fig 2). Distinct patterns of intrinsic spectral-power associations were observed on these sub-second and seconds timescales, characterized by region—anterior, lateral, central, versus posterior—and spatial scale—within versus between EEG-derived current sources. While future research is necessary to understand the functional implications of these intrinsic spectral-power association patterns on perceptual, attentional and cognitive processes, we consider some straightforward interpretations.

The overall results suggest that the fast sub-second-timescale coordination of spectral power is limited to local amplitude modulation and insulated within-frequency long-distance interactions, while the characteristic columnar patterns of cross-frequency interactions emerge on the slower seconds timescale.

In particular, the sub-second associations for the $\beta$-$\gamma$ frequencies at $\Delta f \sim$ 10 Hz at posterior sites (e.g., Fig 3A) suggest that the $\beta$-$\gamma$ oscillations, implicated in a variety of attentional and cognitive processes (e.g., [7,10,11–13,15,44]), are amplitude-modulated at ~10 Hz in the occipital region. In general, the detection of consistent co-occurrences of $f_1$ Hz and $f_2$ Hz oscillations at a single source implies the presence of a $(f_1+f_2)/2$ Hz oscillation being amplitude-modulated at $|f_2–f_1|$ Hz. Thus, the consistent co-occurrences of the $\beta$-$\gamma$ oscillations at $\Delta f \sim$ 10 Hz at posterior sites suggest that the $\beta$-$\gamma$ oscillations are amplitude-modulated at ~10 Hz in the occipital region. Amplitude-modulation at ~10 Hz may in turn imply that the amplitude dynamics of the $\beta$-$\gamma$ oscillations may be phase-coupled to $\alpha$-band (8–12 Hz) oscillations in the occipital region. Indeed, a human MEG study reported that $\gamma$ bnad (30–70 Hz) amplitude variations

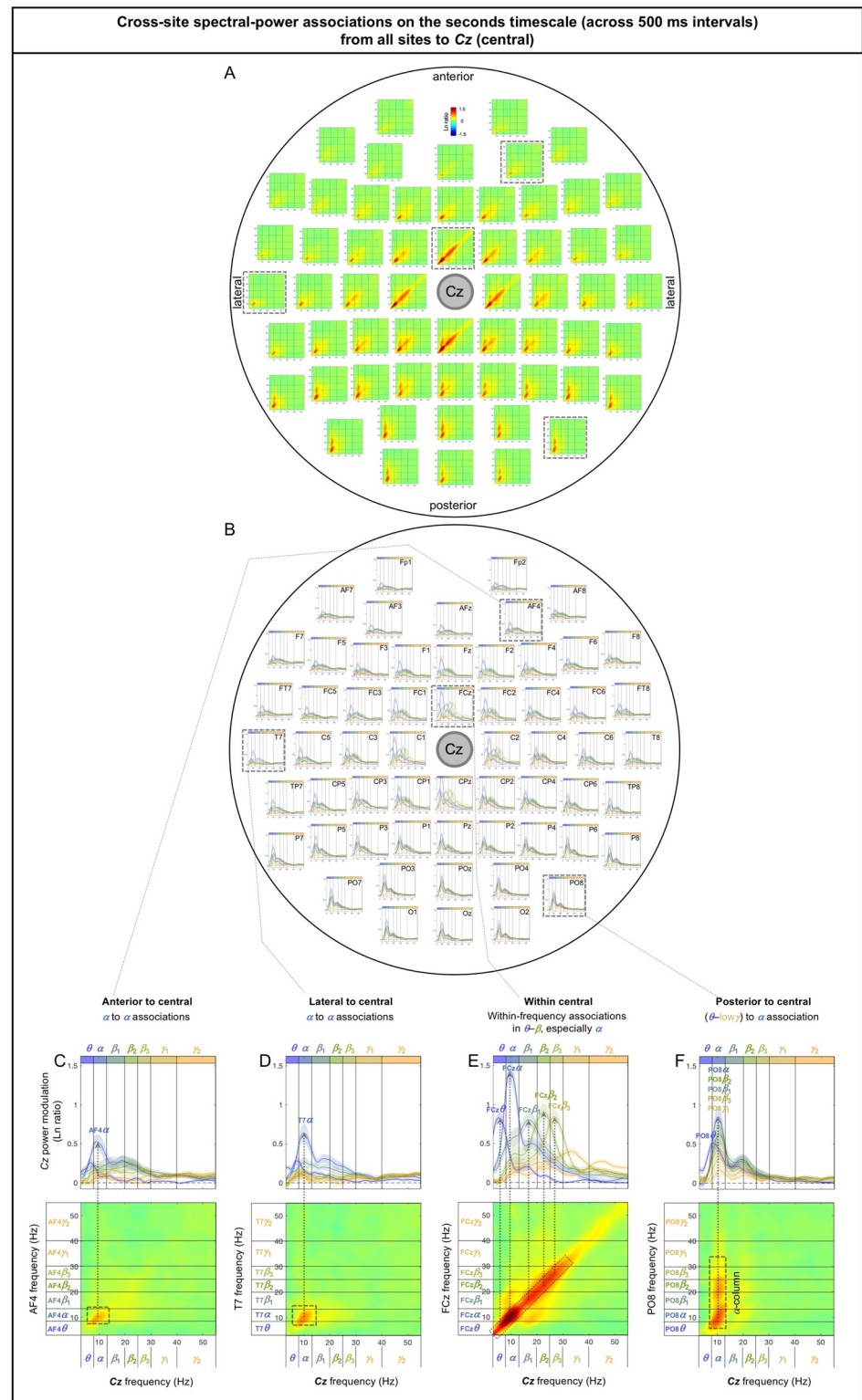

**Fig 12. Cross-site spectral-power associations on the seconds timescale (across 500 ms intervals) from all sites to Cz (a representative central site). A.** 2D spectral-power association plots. Some associations are columnar, indicating multiple frequencies associated with a specific (primarily *α*) band, and some are diagonal, indicating within-frequency associations. **B.** Replotting of A. Each curve shows the spectral-power modulation at *Cz* associated with the probe-site power variation in each major frequency band (lower-to-higher frequency bands labeled with cooler-to-warmer

colors), with the shaded regions indicating ±1 standard error of the mean (with participants as the random effect). In other words, each curve shows a horizontal slice of a 2D-association plot for a specific probe-site frequency band (see the rectangular boxes in the bottom panels). **C-F.** Representative associations. **C.** *AF4* (an anterior site) to *Cz* spectral-power associations, characterized by the $\alpha$ band at *AF4* associated with the $\alpha$ band at *Cz*. **D.** *T7* (lateral) to *Cz* spectral-power associations, characterized by the $\alpha$ band at *T7* associated with the $\alpha$ band at *Cz* (as in C). **E.** *FCz* (an adjacent site) to *Cz* spectral-power associations, characterized by the diagonal (within-frequency) associations in the $\theta$-$\boldsymbol{\beta}$ (especially $\boldsymbol{\alpha}$) bands. **F.** *PO8* (a posterior site) to *Cz* spectral-power associations, characterized by the $\theta$-low$\gamma$ bands at *PO8* associated with the $\alpha$ band at *Cz*—$\boldsymbol{\alpha}$-column. Note that spectral-power associations from most posterior sites to *Cz* are characterized by $\boldsymbol{\alpha}$-columns, that is, the $\theta$-low$\gamma$ bands at most posterior sites were associated with the $\boldsymbol{\alpha}$ band at *Cz* (see the posterior region of A and B). The color scale is shown at the top of part A.

were coupled to the phase of $\alpha$ (8-13Hz) oscillations primarily in the occipital area during rest with the eyes closed [17]. A monkey unit-recording study also reported that $\gamma$ band (30-200Hz) amplitude variations were coupled to the phase of $\alpha$ (7-14Hz) oscillations in V1 during rest [45]. These results taken together suggest that occipital $\alpha$ oscillations play a role in organizing $\beta$-$\gamma$ oscillations into $\alpha$-band amplitude-modulated (AM) packets. If so, $\alpha$ power may covary with $\beta$-$\gamma$ power in the posterior region because extensively synchronized (high power) $\alpha$ oscillations may be required to organize extensively synchronized (high power) $\beta$-$\gamma$ oscillations into $\alpha$-band AM packets.

We did not find consistent posterior associations between the $\alpha$ and $\beta$-$\gamma$ frequencies on the sub-second timescale. This may not be surprising. The fact that the $\beta$-$\gamma$ associations were primarily confined along the 45° lines at $\Delta f \sim$ 10 Hz (e.g., Fig 3A) suggests that the $\beta$-$\gamma$ frequency oscillations were amplitude-modulated at $\Delta f \sim$ 10 Hz relatively independently of one another; otherwise, the spectral power across the $\beta$-$\gamma$ frequencies would have been broadly associated. This means that the $\alpha$-band AM packets for different frequency pairs would have been temporally dispersed within a 500 ms interval. Therefore, even if the $\alpha$ oscillations played a role in organizing the $\beta$-$\gamma$ oscillations into $\alpha$-band AM packets, the $\alpha$ power would not necessarily fluctuate with the $\beta$-$\gamma$ power on the sub-second timescale. For example, within any given sub-second interval, the $\alpha$ power would have likely been consistently high when the various $\beta$-$\gamma$ oscillations (to be organized into $\alpha$-band AM packets) were elevated and consistently low when the various $\beta$-$\gamma$ oscillations were reduced. This reasoning predicts that the associations between the $\alpha$ power and the $\beta$-$\gamma$ power should be observed on a longer timescale. Indeed, on the seconds timescale, posterior sites yielded vertical columns at the $\alpha$ band—$\alpha$-columns, indicating that the $\alpha$ power co-varied with the $\beta$-low$\gamma$ power over the slower timescale. Thus, the $\Delta f \sim$ 10 Hz characteristic of the sub-second-timescale associations and the $\alpha$-column characteristic of the seconds-timescale associations at posterior sites jointly suggest that occipital $\alpha$ oscillations play a role in organizing $\beta$-$\gamma$ oscillations into $\alpha$-band AM packets.

This interpretation raises the possibility that the posterior processes mediated by $\beta$-$\gamma$ oscillations may be transmitted to other regions as $\alpha$-band AM packets. If so, the posterior $\beta$-$\gamma$ oscillations may generate $\alpha$ oscillations at the receiving neural populations through non-linear interactions involving rectification. Examples of such rectifying interactions have been seen as binocular beats (e.g., [46]), heard as binaural beats (e.g., [47]), and observed in visual and auditory evoked EEG potentials (e.g., [48,49]). Thus, the $\beta$-$\gamma$ power variations organized into $\alpha$-band AM packets at posterior sites may be associated with $\alpha$ power variations at other sites. Our results support this prediction. The cross-site spectral-power associations from posterior sites to other sites were consistently characterized by $\alpha$-columns (e.g., Fig 10A–10D; also see the posterior region of Figs 11–14), indicating that the $\theta$-low$\gamma$ power (including the $\beta$-$\gamma$ power) variations at posterior sites selectively coincided with the $\alpha$ power modulation at other sites on the seconds timescale. The fact that these associations were columnar rather than square-shaped (which would have indicated broad associations across the $\beta$-$\gamma$ frequencies) is

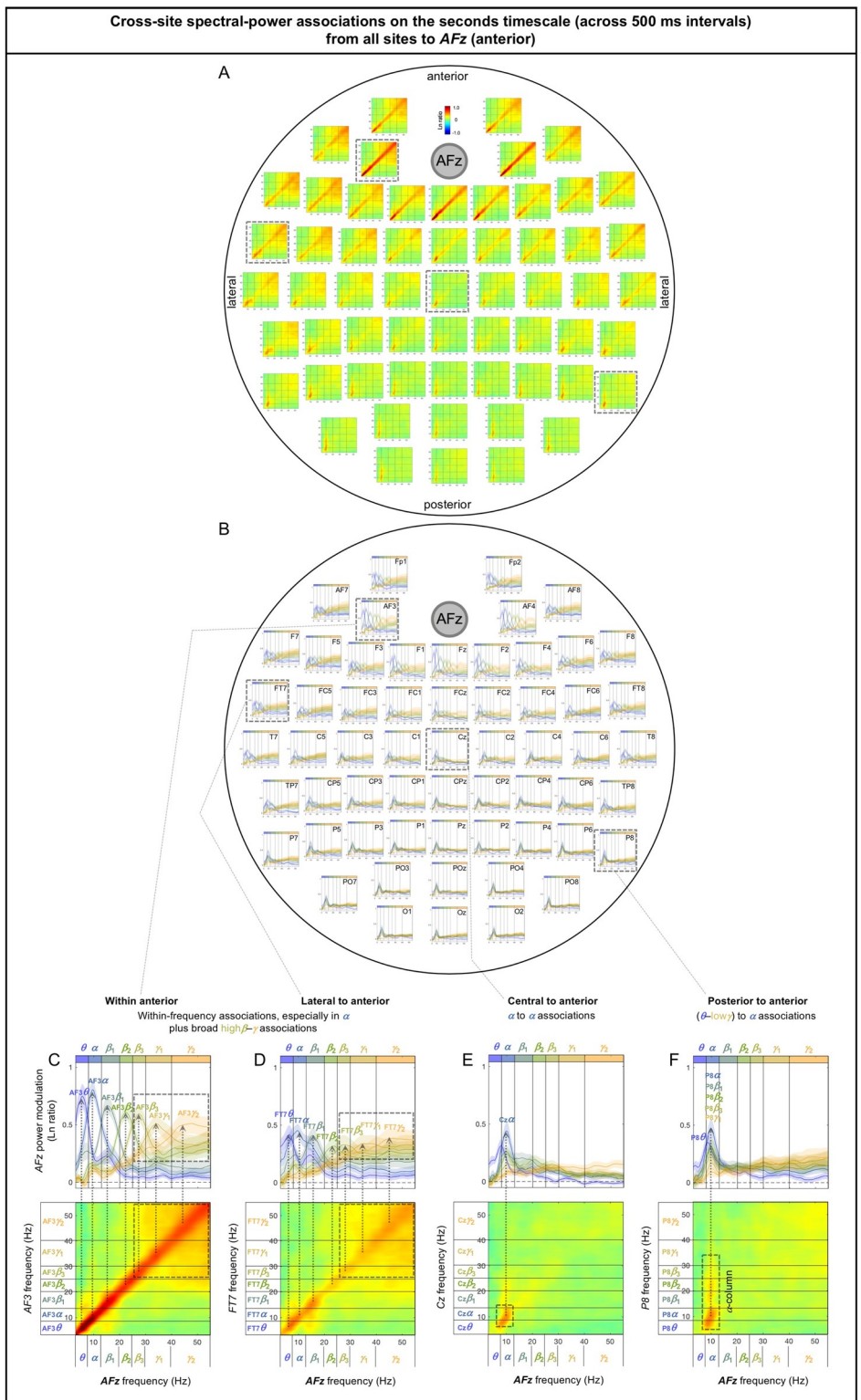

**Fig 13. Cross-site spectral-power associations on the seconds timescale (across 500 ms intervals) from all sites to AFz (a representative central site). A.** 2D spectral-power association plots. Some associations are columnar, indicating multiple frequencies associated with a specific (primarily *α*) band, some are square, indicating broad associations among multiple frequencies, and some are diagonal, indicating within-frequency associations. **B.** Replotting of A. Each curve shows the spectral-power modulation at *AFz* associated with the probe-site power

variation in each major frequency band (lower-to-higher frequency bands labeled with cooler-to-warmer colors), with the shaded regions indicating ±1 standard error of the mean (with participants as the random effect). In other words, each curve shows a horizontal slice of a 2D-association plot for a specific probe-site frequency band (see the rectangular boxes in the bottom panels). **C-F.** Representative associations. **C.** *AF3* (an adjacent anterior site) to *AFz* spectral-power associations, characterized by the diagonal (within-frequency) associations (especially in the $\alpha$ band) as well as the broad associations among the high**$\beta$-$\gamma$** bands. **D.** *T7* (lateral) to *AFz* spectral-power associations, characterized by the diagonal (within-frequency) associations (especially in the $\alpha$ band) as well as the broad associations among the high**$\beta$-$\gamma$** bands (as in C, but weaker). **E.** *Cz* (a central site) to *AFz* spectral-power associations, characterized by the $\alpha$ band at *Cz* associated with the $\alpha$ band at *AFz*. **F.** *P8* (a posterior site) to *AFz* spectral-power associations, characterized by the $\theta$-low$\gamma$ bands at *P8* associated with the $\alpha$ band at *AFz*—$\alpha$-column. Note that spectral-power associations from most posterior sites to *AFz* are characterized by $\alpha$-columns, that is, the $\theta$-low$\gamma$ bands at most posterior sites were associated with the $\alpha$ band at *AFz* (see the posterior region of A and B). The color scale is shown at the top of part A.

consistent with the above interpretation that different $\beta$-$\gamma$ frequencies in the posterior region are relatively independently organized into $\alpha$-band AM packets. Thus, the occipital operations involving $\beta$-$\gamma$ oscillations may be frequency-wise organized into $\alpha$-band AM packets to coordinate their interactions with other regions.

This interpretation is consistent with prior results suggesting that visual perception, awareness, and/or attention oscillate in the $\alpha$ rhythm (e.g., [3,50]). For example, stimulus visibility has been shown to correlate with the specific phase of the posterior EEG $\alpha$ oscillations (e.g., [2], but see [6]). Our results suggest that this correlation between visibility and the posterior $\alpha$ oscillations may be partly mediated by the $\alpha$-band modulation of the information carried by $\beta$-$\gamma$ band oscillations. This possibility is consistent with the proposal that the occipital $\alpha$ rhythms may contribute to periodically organizing early spatially-parallel visual signals, carried by the $\gamma$-band oscillations, into strength-based temporal code; that is, the gradual lifting of the inhibitory phase within each $\alpha$-oscillation cycle may allow stronger signals to overcome the inhibition earlier, resulting in the strength-based temporal sequencing of the spatially-parallel visual sensory signals within each $\alpha$-oscillation cycle (e.g., [9]). This process, which may be detected as $\alpha$-band amplitude-modulation of $\gamma$-band power (as in the current results), may facilitate the serial attending and recognition of objects in downstream visual processing.

So far, we have inferred occipital $\alpha$-phase-to-$\beta$-$\gamma$-amplitude coupling based on the characteristic $\Delta f \sim$ 10 Hz spectral-power associations for the $\beta$-$\gamma$ frequencies on the sub-second timescale (e.g., Fig 3A). Additionally, we observed the $\Delta f \sim$ 3 Hz spectral-power associations for the $\theta$-$\alpha$-frequencies at posterior sites (e.g., Fig 3A). Does this imply $\delta$-phase-to-$\theta$-$\alpha$-amplitude coupling in the posterior region? We also observed the $\Delta f \sim$ 16 Hz associations for the $\gamma$ frequencies at lateral sites (e.g., Fig 3B). Does this imply low$\beta$-phase-to-$\gamma$-amplitude coupling in the lateral region?

We sought corroborating evidence for the relationship between the $\Delta f$ associations and phase-amplitude coupling. Our strategy was to determine whether the types of phase-power coupling implied by the $\Delta f$ spectral-power associations matched the phase-amplitude coupling assessed with a standard method of computing the Modulation Index, *MI* (e.g., [51,52]). In short, the mean amplitude of the faster frequency band—amplitude frequency or *Af*—that coincides with each phase bin of the slower frequency band—phase frequency or *Pf*—is computed (over some temporal interval) to determine the degree to which the distribution of the *Af* amplitude over the *Pf* phase is peaky (i.e., the degree to which large *Af* amplitudes are concentrated at specific phase(s) of the *Pf* oscillation), using an entropy measure [51]. The specific equation of *MI* yields a value of 0 if the distribution is flat and 1 if the distribution is maximally peaky (i.e., if the *Af* amplitudes are non-zero only at a single *Pf* phase bin). We used the full data length (~300 s) to compute *MI*'s, and also subtracted the control *MI*'s computed with the corresponding phase-scrambled data (per site per participant).

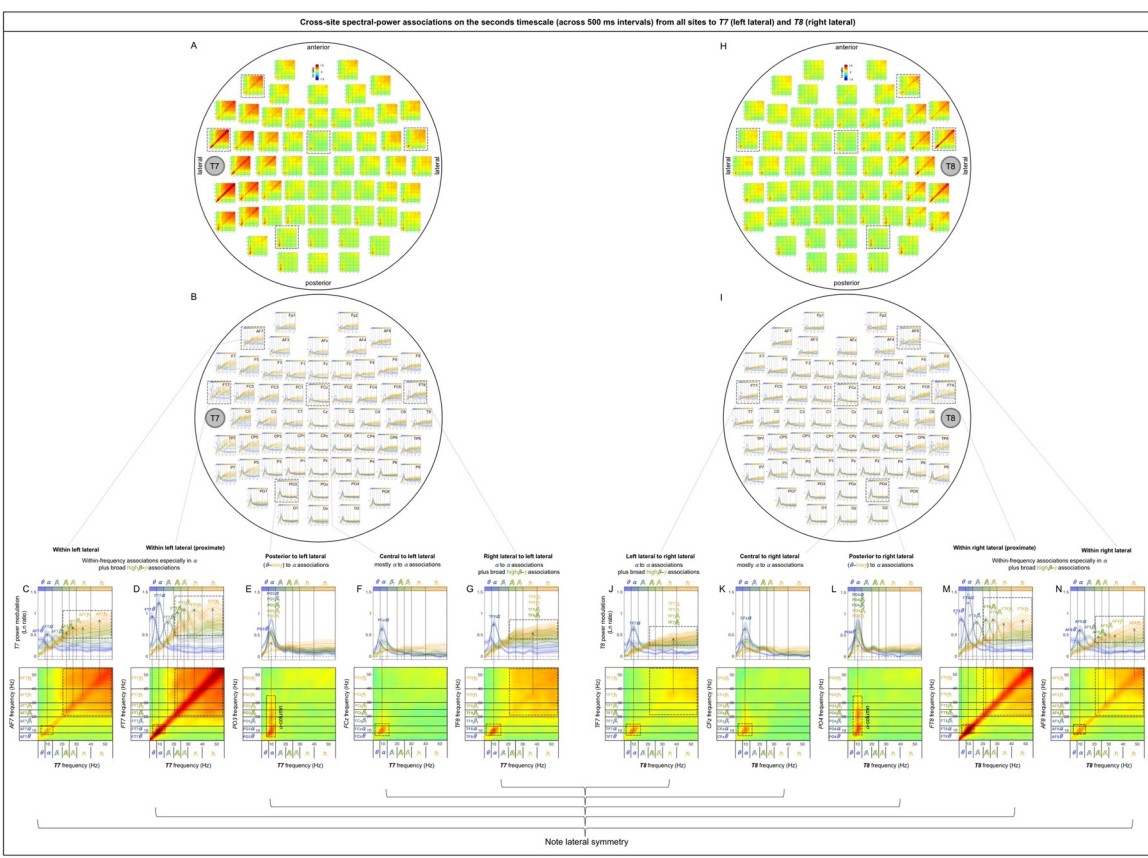

**Fig 14. Cross-site spectral-power associations on the seconds timescale (across 500 ms intervals) from all sites to *T7* (a representative left lateral site) and *T8* (a representative right lateral site). A, H.** 2D spectral-power association plots for *T7* and *T8*. Some associations are columnar, indicating multiple frequencies associated with a specific (primarily **α**) band, some are square, indicating broad associations among multiple frequencies, and some are diagonal, indicating within-frequency associations. **B, I.** Replotting of A. Each curve shows the spectral-power modulation at *T7* or *T8* associated with the probe-site power variation in each major frequency band (lower-to-higher frequency bands labeled with cooler-to-warmer colors), with the shaded regions indicating ±1 standard error of the mean (with participants as the random effect). In other words, each curve shows a horizontal slice of a 2D-association plot for a specific probe-site frequency band (see the rectangular boxes in the bottom panels). **C-G, J-N.** Representative associations for *T7* and *T8*. Note the bilateral symmetry. **C, N.** *AF7/AF8* (ipsilateral anterior sites) to *T7/T8* spectral-power associations, characterized by the diagonal (within-frequency) associations (especially in the **α** band) as well as the broad associations among the high**β**-**γ** bands. **D, M.** *FT7/FT8* (adjacent sites) to *T7/T8* spectral-power associations, characterized by the diagonal (within-frequency) associations (especially in the **α** band) as well as the broad associations among the high**β**-**γ** bands (similar to C, N, but stronger). **E, L.** *PO3/PO4* (posterior sites) to *T7/T8* spectral-power associations, characterized by the **θ**-low**γ** band at *PO3/PO4* associated with the **α** band at *T7/T8*—**α**-column. **F, K.** *FCz* (a central site) to *T7/T8* spectral-power associations, characterized by the **α** band at *FCz* associated with the **α**-band at *T7/T8*. **G, J.** *TF8/TF7* (contralateral sites) to *T7/T8* associations, characterized by the **α** band at *TF8/TF7* associated with the **α** band at *T7/T8* as well as the broad associations among the high**β**-**γ** bands. Note that spectral-power associations from most posterior sites to *T7/T8* are characterized by **α**-columns, that is, the **θ**-low**γ** bands at most posterior sites were associated with the **α** band at *T7/T8* (see the posterior region of A, B and H, I). The color scale is shown at the top of part A and H.

The sub-second $\Delta f$~ 3 Hz associations for the $\theta$-$\alpha$-frequencies at posterior sites (e.g., Fig 3A) potentially imply a selective coupling of the $\theta$-$\alpha$ amplitudes to the phase of the $\delta$-band oscillations. However, we did not observe that the *MI*'s for the *Pf* of 3–4 Hz was selectively elevated for the *Af* of $\theta$-$\alpha$ bands at the corresponding sites (data not shown). Thus, it is inconclusive as to whether the $\Delta f$~ 3 Hz associations at posterior sites imply phase-amplitude coupling.

The results were more encouraging for the $\Delta f$~ 10 Hz and $\Delta f$~ 16 Hz associations. The $\Delta f$~ 10 Hz associations for the $\beta$-$\gamma$-frequencies at posterior sites potentially imply a coupling of $\beta$-$\gamma$ amplitudes to the phase of the $\alpha$-band oscillations (a representative example from *PO8* shown

in Fig 15B). This would selectively elevate *MI* for the combinations of *Pf* = 8–12 Hz (*α* band) with *Af* = 15–30 Hz (*β* band) and *Af* = 30–55 Hz (*γ* band). This prediction was confirmed (representative results from *PO8* shown in Fig 15C; see the data points highlighted in the dashed rectangle). To confirm selectivity, *MI* for the combination of *Pf* = 13–17 Hz (low*β* band) with neither *Af* = 15–30 Hz (*β* band) nor *Af* = 30–55 Hz (*γ* band) was elevated (see non-highlighted data points in Fig 15C). Similarly, the *Δf* ~ 16 Hz associations for the *γ*-frequencies at lateral sites (e.g., Fig 3B) potentially imply a coupling of the *γ* amplitudes to the phase of the low*β*-band oscillations (a representative example from *C6* shown in Fig 15E). This would selectively elevate *MI* for the combination of *Pf* = 13–17 Hz (low*β* band) with *Af* = 30–55 Hz (*γ* band). This prediction was confirmed (representative results from *C6* shown in Fig 15F; see the data point highlighted in the dashed rectangle). To confirm selectivity, *MI* for the combination of *Pf* = 13–17 Hz (low*β* band) with *Af* = 15–30 Hz (*β* band) was not elevated; neither was *MI*

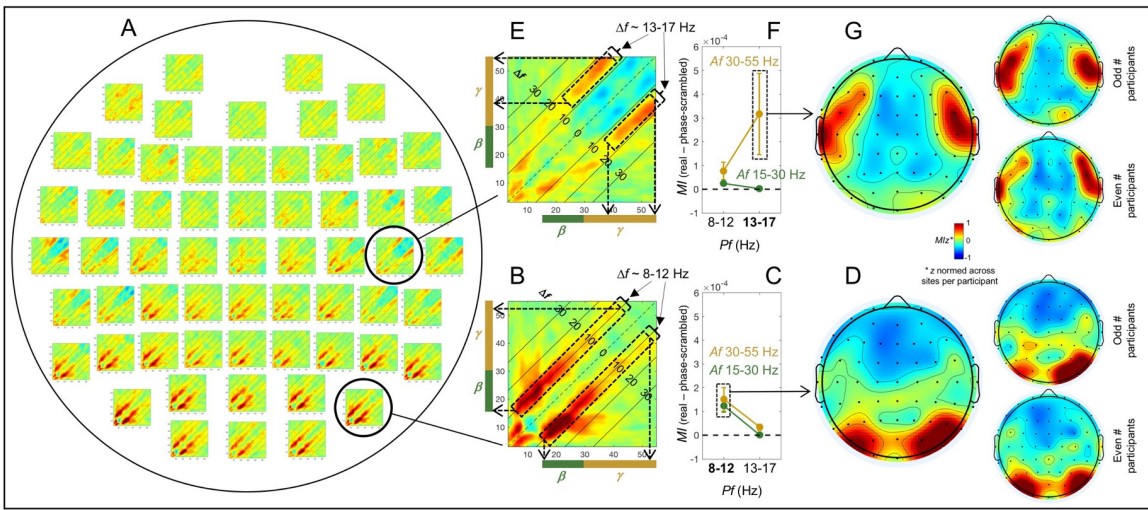

**Fig 15. Relationship between the posterior *Δf* ~ 10 Hz and lateral *Δf* ~ 16 Hz spectral-power associations on the sub-second timescale and phase-amplitude coupling assessed as the Modulation Index (*MI*). A.** The spatial distribution of the spectral-power associations on the sub-second timescale (same as Fig 3D). **B.** A representative example of the posterior *Δf* ~ 10 Hz (8–12 Hz, *α* band) associations for the *β*-*γ* frequencies at *PO8*. This pattern of spectral-power associations may imply a phase-amplitude coupling of the phase-frequency band (*Pf*) corresponding to *Δf* (8–12 Hz, *α* band) with the amplitude-frequency band (*Af*) in both the *β* (green bars) and *γ* (gold bars) ranges (see dashed rectangles and arrows). **C.** The phase-amplitude coupling assessed as *MI* (*MI* for real data minus *MI* for phase-scrambled data) at *PO8*. As implied by the spectral-power associations shown in B, *MI* was elevated selectively for the combination of *Pf* = 8–12 Hz (*α* band) with both *Af* = 15–30 Hz (*β* band) and *Af* = 30–55 Hz (*γ* band) (highlighted with the dashed rectangle). To confirm selectivity, *MI* was not elevated for the combination of *Pf* = 13–17 (low*β* band) with *Af* = 15–30 Hz (*β* band) or *Af* = 30–55 Hz (*γ* band). The error bars represent ±1 standard error of the mean with participants as the random effect. **D. Left.** Spatial distribution of *MIz* (*z*-normalized across sites per participant to focus on the spatial patterns of *MI*) averaged for the combinations of *Pf* = 8–12 Hz (*α* band) with *Af* = 15–30 Hz (*β* band) and *Af* = 30–55 Hz (*γ* band) (e.g., the dashed rectangle in C) predicted to be elevated by the posterior spectral-power associations shown in B. The posterior distribution of *MIz* largely parallels the posterior distribution of the *Δf* ~ 8–12 Hz associations (see A). **Right.** The *MIz* distributions plotted separately for the odd and even numbered participants showing the degree of inter-participant variability. **E.** A representative example of the lateral *Δf* ~ 16 Hz (13–17 Hz, low*β* band) associations for the *γ* frequencies at *C6*. This pattern of spectral-power associations may imply a phase-amplitude coupling of the phase-frequency band (*Pf*) corresponding to *Δf* (13–17 Hz, low*β* band) with the amplitude-frequency band (*Af*) in the *γ* (gold bars) range (see the dashed rectangles and arrows). **F.** The phase-amplitude coupling assessed as *MI* (*MI* for real data minus *MI* for phase-scrambled data) at *C6*. As implied by the spectral-power associations shown in E, *MI* was elevated selectively for the combination of *Pf* = 13–17 Hz (low*β* band) with *Af* = 30–55 Hz (*γ* band) (highlighted with dashed rectangle). To confirm selectivity, *MI* was not elevated for the combination of *Pf* = 13–17 (low*β* band) with *Af* = 15–30 Hz (*β* band); neither was it elevated for the combination of *Pf* = 8–12 Hz (*α* band) with *Af* = 15–30 Hz (*β* band) or *Af* = 30–55 Hz (*γ* band). The error bars represent ±1 standard error of the mean with participants as the random effect. **G. Left.** Spatial distribution of *MIz* (*z*-normalized across sites per participant to focus on the spatial patterns of *MI*) for the combination of *Pf* = 13–17 Hz (low*β* band) with *Af* = 30–55 Hz (*γ* band) (e.g., the dashed rectangle in F) predicted to be elevated by the lateral spectral-power associations shown in E. The lateral distribution of *MIz* largely parallels the lateral distribution of the *Δf* ~ 13–17 Hz associations (see A). **Right.** The *MIz* distributions plotted separately for the odd and even numbered participants showing the degree of inter-participant variability.

elevated for the combination of *Pf* = 8–12 Hz (*α* band) with *Af* = 15–30 Hz (*β* band) or *Af* = 30–55 Hz (*γ* band) (see non-highlighted data points in Fig 15F).

We compared the spatial distributions of the *Δf*~ 10 Hz and *Δf*~ 16 Hz spectral-power associations (Fig 15A) with the spatial distributions of *MI* values for the expected phase-power coupling. To this end, we plotted the spatial distribution of the average of the *MI* values for the combinations of *Pf* = 8–12 Hz (*α* band) with *Af* = 15–30 Hz (*β* band) and *Af* = 30–55 Hz (*γ* band), predicted to be elevated in relation to the *Δf*~ 10 Hz spectral-power associations (e.g., Fig 15B–15C). These *MI* values were first *z*-normalized across sites per participant (*MIz*) so that the analysis focused on the spatial patterns of *MI*. The distribution of *MIz* covered the posterior region (Fig 15D, left) largely paralleling the posterior distribution of the *Δf*~ 10 Hz association (Fig 15A). The posterior *MIz* distribution appears to be reasonably reliable as the patterns obtained from the odd and even numbered participants were similar (Fig 15D, right). Similarly, we plotted the spatial distribution of *MIz* for the combination of *Pf* = 13–17 Hz (low*β* band) with *Af* = 30–55 Hz (*γ* band), predicted to be elevated in relation to the *Δf*~ 16 Hz spectral-power associations (e.g., Fig 15E–15F). The distribution covered the lateral region (Fig 15G, left) largely paralleling the lateral distribution of the *Δf*~ 16 Hz association (Fig 15A). Again, the lateral *MIz* distribution appears to be reasonably reliable as the patterns obtained from the odd and even numbered participants were similar (Fig 15G, right).

These results provide indirect evidence suggesting that at least the spectral-power associations at *Δf*~ 10 Hz and *Δf*~ 16 Hz on the sub-second timescale are related to amplitude-modulation of higher-frequency oscillations by the phase of *Δf* oscillations. For the posterior *Δf*~ 10 Hz associations on the sub-second timescale that imply *α*-phase-to-*β*-*γ*-amplitude coupling, our results suggest a coherent interpretation relating them to the *α*-column associations on the seconds timescale observed at posterior sites and between posterior and non-posterior sites (see above). However, the current results do not provide a coherent interpretation for the lateral *Δf*~ 16 Hz associations that imply low*β*-phase-to-*γ*-amplitude coupling.

Explanations are also required as to why the (fast) sub-second-timescale cross-site (long-distance) diagonal associations were dominated by *α* (8-15hz), *β*-*γ* (>30Hz), or a combination of *α* and *β*-*γ* frequencies, but never dominated by *β* (15-30Hz) frequencies (e.g., Fig 5; also see Figs 6–9). Given that diagonal associations indicate insulated within-frequency interactions (i.e., independent interactions for different frequencies), the fast, long-distance diagonal associations may reflect feedforward (dominated by frequencies above ~40 Hz; [44]) and feedback (dominated by 5–20 Hz while peaking at ~10Hz; [44]) interactions. Note that, although our participants rested with their eyes closed, they continuously received and processed environmental sensory stimuli. The fact that the fast diagonal long-distance associations to *Cz* (a representative central site) were never dominated by the *β*-*γ* frequencies (Fig 7) may potentially suggest that central sites do not strongly engage in feedforward interactions while people rest with their eyes closed.

The cross-site associations generally diminished with distance. However, there were cases where strong associations persisted between distant sites (with weaker associations at intermediate distances). Those "super" long-distance associations were typically *α*-band to *α*-band on both the sub-second and seconds timescales, but there were some exceptions, including the fast within-frequency *β*-*γ* associations from lateral sites, *T7*/*T8*, to the inferior posterior site, *Oz*, and vice versa, on the sub-second timescale (e.g., Figs 6C, Fig 9F and 9K), and the slower broad *β*-*γ* associations from distant contralateral sites such as *TF8*/*TF7* to lateral sites, *T7*/*T8*, on the seconds timescale (e.g., Fig 14G and 4J; also see Fig 14A–14B and 14H–14I). The former may be relevant to hierarchical visual processing occurring on the (fast) sub-second timescale whereas the latter may be relevant to cross-hemisphere coordination occurring on the (slower) seconds timescale.

When analyzing cross-frequency associations using frequency decomposition methods, one needs to be mindful of the artifactual harmonics generated by non-sinusoidal waveforms. In particular, recent computational studies have demonstrated that non-sinusoidal waveforms of $\alpha$ oscillations could generate harmonics that mimic the coupled $\beta$-$\gamma$ oscillations (e.g., [53,54]). This waveform artifact could contribute broad spectral-power associations among the $\alpha$ through $\beta$-$\gamma$ frequencies as the power of artifactual $\beta$-$\gamma$ harmonics would synchronously covary with the power of the non-sinusoidal $\alpha$ oscillations. Nevertheless, because the signals at each EEG current-source (site) reflect extensive spatial summation of neuronal signals at variable delays, any observable effects of artifactual spectral-power associations due to the harmonics generated by non-sinusoidal waveforms would have been attenuated [55]. Further, most of the characteristic spectral-power associations we obtained on the sub-second and seconds timescales are not characterized by broad associations among the $\alpha$ through $\beta$-$\gamma$ frequencies.

Finally, the distinct patterns of intrinsic spectral-power associations characterized by the parallel-to-diagonal, diagonal, columnar, and square associations may provide a useful reference for understanding how intrinsic global dynamics adjust to sensory dynamics and behavioral demands. For example, we have obtained some preliminary results suggesting that, while people watched a silent nature video, the intrinsic posterior $\Delta f \sim$ 10Hz associations for the $\beta$-$\gamma$ power on the sub-second timescale (obtained while participants rested with their eyes closed) strengthened but $\Delta f$ shifted to the lower$\alpha$/upper$\theta$ frequencies (leaving the lateral $\Delta f \sim$ 16Hz associations for the $\gamma$ frequencies unchanged), while the $\alpha$-column associations from posterior sites to other sites on the seconds timescale disappeared (Menceloglu, Grabowecky, & Suzuki, unpublished data). While these preliminary results suggest that intrinsic spectral-power associations undergo characteristic adjustments from resting with the eyes closed to free viewing, they need to be confirmed and the sources of the adjustments investigated, for example, by determining whether they are due to the eyes being open per se, the presence of visual stimulation, and/or the act of free viewing. Investigating such adjustments in relation to various sensory dynamics and behavioral demands may contribute to an understanding of how internal neural dynamics are optimized for interacting with environmental dynamics.

In summary, we examined the global distribution of spectral-power associations, focusing on both the local characteristics reflected in the cross-frequency associations within individual EEG-derived current sources (sites) and the long-distance characteristics reflected in the cross-frequency associations between sites. Distinct association patterns emerged on the (fast) sub-second and (slower) seconds timescales, suggesting that the fast coordination of spectral power appears to be limited to local amplitude modulation (suggestive of phase-power coupling) and insulated within-frequency long-distance interactions (dominated by the $\alpha$ and $\gamma$ bands, likely reflecting feedforward and feedback processes), while the characteristic columnar patterns of cross-frequency interactions emerge on the slower timescale. While the functional significance of these intrinsic spectral-power associations awaits future research, the patterns of associations within the posterior region and between the posterior and other regions suggest that the intrinsic occipital dynamics are structured in such a way that the $\beta$-$\gamma$ oscillations are organized into $\alpha$-band (~10 Hz) amplitude-modulated packets to interact with other regions.

## Supporting information

**S1 Fig. Inter-participant consistency of the within-site spectral-power associations on the sub-second and seconds timescales.**
(JPEG)

**S2 Fig. Inter-participant consistency of the cross-site spectral-power associations on the sub-second timescale (within a 500 ms interval) from all sites to representative posterior**

(*Oz*), central (*Cz*), anterior (*AFz*), and lateral (*T7/T8*) sites.
(JPEG)

**S3 Fig. Inter-participant consistency of the cross-site spectral-power associations on the seconds timescale (across 500 ms intervals) from all sites to representative posterior (*Oz*), central (*Cz*), anterior (*AFz*), and lateral (*T7/T8*) sites.**
(JPEG)

## Author Contributions

**Conceptualization:** Satoru Suzuki.

**Data curation:** Melisa Menceloglu.

**Formal analysis:** Satoru Suzuki.

**Funding acquisition:** Melisa Menceloglu.

**Investigation:** Melisa Menceloglu, Marcia Grabowecky, Satoru Suzuki.

**Methodology:** Melisa Menceloglu, Satoru Suzuki.

**Project administration:** Melisa Menceloglu, Marcia Grabowecky, Satoru Suzuki.

**Resources:** Melisa Menceloglu, Marcia Grabowecky, Satoru Suzuki.

**Software:** Satoru Suzuki.

**Supervision:** Marcia Grabowecky, Satoru Suzuki.

**Validation:** Satoru Suzuki.

**Visualization:** Melisa Menceloglu, Satoru Suzuki.

**Writing – original draft:** Satoru Suzuki.

**Writing – review & editing:** Melisa Menceloglu, Marcia Grabowecky, Satoru Suzuki.

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
