## [Decision Letter · Decision Letter 0]

9 Mar 2020

PONE-D-20-00985

Spectral-power associations reflect amplitude modulation and within-frequency interactions on the sub-second timescale and cross-frequency interactions on the seconds timescale

PLOS ONE

Dear Suzuki,

Thank you for submitting your manuscript to PLOS ONE. After careful consideration, we feel that it has merit but does not fully meet PLOS ONE’s publication criteria as it currently stands. Therefore, we invite you to submit a revised version of the manuscript that addresses the points raised during the review process.

We would appreciate receiving your revised manuscript by Apr 23 2020 11:59PM. To enhance the reproducibility of your results, we recommend that if applicable you deposit your laboratory protocols in protocols.io, where a protocol can be assigned its own identifier (DOI) such that it can be cited independently in the future. For instructions see: http://journals.plos.org/plosone/s/submission-guidelines#loc-laboratory-protocols

We look forward to receiving your revised manuscript.

Kind regards,

Vladyslav Vyazovskiy, PhD

Academic Editor

PLOS ONE

Journal Requirements:

2. Please provide additional details regarding participant consent. In the Methods section, please ensure that you have specified what type of consent you obtained (for instance, written or verbal) and whether the ethics committee approved this consent procedure. If verbal consent was obtained please state why it was not possible to obtain written consent and how verbal consent was recorded. If your study included minors, state whether you obtained consent from parents or guardians.

Reviewers' comments:

Reviewer's Responses to Questions

**Comments to the Author**

1. Is the manuscript technically sound, and do the data support the conclusions?

Reviewer #1: Partly

Reviewer #2: Yes

2. Has the statistical analysis been performed appropriately and rigorously? 

Reviewer #1: Yes

Reviewer #2: N/A

3. Have the authors made all data underlying the findings in their manuscript fully available?

Reviewer #1: No

Reviewer #2: No

4. Is the manuscript presented in an intelligible fashion and written in standard English?

Reviewer #1: Yes

Reviewer #2: Yes

5. Review Comments to the Author

Reviewer #1: In this article, the authors present an original method to quantify spectral-power associations and used it to examine the intra- and inter-sites distribution of spectral-power associations in EEG recordings of subjects at rest. Two independent analysis were implemented for fast (sub-second) and slow (seconds) timescales. The authors also reflect about the relevance of their findings and provide some straightforward interpretations. The paper is therefore of relevance in the field of cross-frequency couplings and the analysis of its functional significance.

The experiments are well conceived and data analysis and statistics appears sound. I believe this paper will be of interest both to specialist in data processing and experimentalists. I would like to recommend the publication of the manuscript after the authors consider the revision of the points detailed in the attached file.

Reviewer #2: In this manuscript, the amplitude modulation and within-frequency interactions on two specific timescales of sub-second and seconds are illustrated using the plots of spectral-power associations. This approach is interesting to me. There are some comments for authors:

Major comments:

1) This approach focuses on the methodology to represent the spectral-power association among the recordings of multi-channels current source density in resting eye-close state. Any new findings with clear physiological meanings? There is no conclusion in this manuscript. Conclusions are important for audiences to catch your contributions and finding in this study.

2) A lot of spectral-power association plots are used to illustrate the sub-second and seconds timescale modulations within single site or cross-site. It is interesting that cross-frequency associations within-site represent the amplitude modulation with Δf and the cross-site associations centralized on the diagonal without Δf. These findings are interesting, but it is difficult for audiences to catch your findings if there is no quantitative statements.

3) Only the cross-site association between one of the referenced sites (AFz, Oz, Cz, T7 & T8) and one out of the other electrodes were presented in this manuscript. There are a total 64*63/2 cross-site associations can be observed in the functional connectivity using 64-channel montage. However, it is impossible to illustrate all cross-site associations in this manuscript. My opinion is to develop a quantitative measure for spectral-power association. The quantitative measure benefits to describing your findings and make it possible to conduct further analysis to functional connectivity for different conditions in future works.

4) Only eye-closed state was investigated in this study, I am interested for what kind within-site and sub-second association (as shown in Figure 3) will be presented in compared with the outcome of eye-close condition. If it possible, authors can provide some statements to describe this.

5) In the second paragraph of page 6 for generating phase-scrambled controls authors mentioned “we chose discrete cosign transform, DCT”. Is this statement correct? The full name of DCT should be “discrete cosine transform”, isn’t it?

6. PLOS authors have the option to publish the peer review history of their article (what does this mean?). If published, this will include your full peer review and any attached files.

Reviewer #1: Yes: Damián Dellavale

Reviewer #2: No

---

## [Author Response · Author response to Decision Letter 0]

21 Apr 2020

Our response to the reviewers' comments has been uploaded as the file labeled Response to Reviewers.

---

## [Editor Report · Decision Letter 1]

27 Apr 2020

Spectral-power associations reflect amplitude modulation and within-frequency interactions on the sub-second timescale and cross-frequency interactions on the seconds timescale

PONE-D-20-00985R1

Dear Dr. Suzuki,

We are pleased to inform you that your manuscript has been judged scientifically suitable for publication and will be formally accepted for publication once it complies with all outstanding technical requirements.

With kind regards,

Vladyslav Vyazovskiy, PhD

Academic Editor

PLOS ONE
---

## [Editor Report · Acceptance letter]

4 May 2020

PONE-D-20-00985R1 

Spectral-power associations reflect amplitude modulation and within-frequency interactions on the sub-second timescale and cross-frequency interactions on the seconds timescale 

Dear Dr. Suzuki:

I am pleased to inform you that your manuscript has been deemed suitable for publication in PLOS ONE. Congratulations! Your manuscript is now with our production department. 

With kind regards,

on behalf of

Dr. Vladyslav Vyazovskiy 

Academic Editor

PLOS ONE